# Contribution of Full Wave Acoustic Logging to the Detection and Prediction of Karstic Bodies

**Jean-Luc MARI** [1,*]**, Gilles POREL** [2] **and Frederick DELAY** [3]

[1] Sorbonne Université, CNRS, EPHE, UMR 7619 METIS, 4 Place Jussieu, 75005 Paris, France
[2] IC2MP UMR 7285, Department of Earth Sciences, Université de Poitiers, CNRS, HydrASA, F-86000 Poitiers, France; gilles.porel@univ-poitiers.fr
[3] LHyGeS–UMR 7517, Department of Earth Sciences, Université de Strasbourg, CNRS, ENGEES, F-67000 Strasbourg, France; fdelay@unistra.fr
* Correspondence: jeanluc90.mari@gmail.com; Tel.: 33-6-41-24-02-25

**Abstract:** A 3D seismic survey was done on a near surface karstic reservoir located at the hydrogeological experimental site (HES) of the University of Poitiers (France). The processing of the 3D data led to obtaining a 3D velocity block in depth. The velocity block was converted in pseudo porosity. The resulting 3D seismic pseudo-porosity block reveals three high-porosity, presumably-water-productive layers, at depths of 30–40, 85–87 and 110–115 m. This paper shows how full wave acoustic logging (FWAL) can be used to validate the results obtained from the 3D seismic survey if the karstic body has a lateral extension over several seismic. If karstic bodies have a small extension, FWAL in open hole can be fruitfully used to: detect highly permeable bodies, thanks to measurements of acoustic energy and attenuation; detect the presence of karstic bodies characterized by a very strong attenuation of the different wave trains and a loss of continuity of acoustic sections; confirm the results obtained by vertical seismic profile (VSP) data. The field example also shows that acoustic attenuation of the total wavefield as well as conversion of downward-going P-wave in Stoneley waves observed on VSP data are strongly correlated with the presence of flow.

**Keywords:** acoustic logging; 3D seismic; vertical seismic profile; Stoneley wave; attenuation; karstic body; porosity; permeability

## 1. Introduction

Many geophysical studies have been undertaken to investigate the presence of karstic structures, to detect conduits and cavities, and to characterize the surrounding host rock.

Electrical resistivity tomography (ERT) is a proven imaging technique well-suited to identifying karst features and partially characterizing karst aquifers [1]: "Karst features can be predominantly air-filled, making them highly resistive (e.g., >1000 $\Omega \cdot$m), or partially or completely water-filled (e.g., 60–1000 $\Omega \cdot$m); in the latter case, depending on the ionic concentration of the groundwater, karst features may have a bulk conductivity ranging from very conductive (e.g., 60–100 $\Omega \cdot$m) to relatively conductive (e.g., 100–250 $\Omega \cdot$m), compared to the host rock(e.g., >2000 $\Omega \cdot$m for not karst limestone). Karst features may also be filled by highly weathered material, such as clays, in which case they will be conductive relative to the host rock (e.g., 60–150 $\Omega \cdot$m)."

The integration of field data from different geophysical methods is the most suitable approach for imaging karstic structures, detecting cavities, and characterizing the host rock [2]. A combination of high-resolution seismic diffraction, reflection, refraction, continuous vertical electric sounding, ground-penetrating radar (GPR) and microgravity methods was used to study the shallow subsurface at a sinkhole-development site on the shore of the Dead Sea in Israel [3]. Microgravimetry and

multichannel analysis of surface waves (MASW) were applied to image karst structures and cavities in the urban environment of Orleans (France) [4]. ERT and GPR were combined to image a 15-m diameter 3-m deep sinkhole in west-central Florida [5]. An example of karstic networks detection showed the interest of combining geophysical measurements involving ERT, magnetic resonance sounding (MRS), "grounded" electrical mapping, and seismic tomography: "1- Seismic and MRS provided the exact location and width of the conduit, to within a few meters; 2- Seismic and electrical data suggested that the limestone medium surrounding the conduit is not homogeneous" [6].

The present paper illustrates the benefit of combining seismic methods (3D refraction tomography, 3D reflection imaging, vertical seismic profile (VSP) and full-wave acoustic logging (FWAL)) for the characterization of a near surface karstic aquifer. The University of Poitiers-France and a few partners triggered the building of the hydrogeological experimental site (HES) near the campus for the single purpose of providing facilities to develop long-term monitoring and experiments that would improve our understanding of flow and transfers in limestone fractured aquifers [7–10].

Thirty-five wells were drilled at the HES to perform hydrogeological experiments, such as slug tests, interference testing, tracer tests, etc. In each well, a borehole televiewer (BHTV) was run. The analysis of the collected optic images was used to detect and visualize karstic bodies. In 2004, a high frequency band (up to 200 Hz) 3D survey was recorded at the HES. The processing of the 3D data resulted in a 3D seismic pseudo-porosity block which revealed three high-porosity layers, at depths of 30–40, 85–87 and 110–115 m. The BHTV images confirmed that the three high-porosity layers detected at the seismic scale were karstic layers.

After a description of the geological context at the HES, the paper briefly reviews the procedure of 3D seismic imaging used to map the distributions of karstic bodies in the near surface karstified subsurface reservoir investigated. Seismic resolution is conventionally estimated as the quarter of the dominant wavelength, ranging here between 8 and 10 m. However, the 3D seismic imaging does not have enough resolution both in the vertical and horizontal directions to detect karstic bodies of small size, like those observed by BHTV in wells M04, MP7 and C1 in the 45–60 m depth interval.

In a second step, it is shown how full wave acoustic logging (FWAL) may contribute to the detection of actual karstic layers and to the prediction of the location of highly permeable water-bearing bodies that could evolve toward karstic bodies. With the idea to detect karstic bodies of small size, FWAL is run in wells MP7 and C1. The acoustic data are recorded in the 1–20 kHz frequency band. If the vertical resolution of the acoustic logging is very high (on the order of 10 cm), the lateral investigation close to the well is limited to a few tens of centimeters.

The results rendered by acoustic logging are confirmed by additional measurements such as flowmeters and VSP with hydrophones in wells.

## 2. Hydrogeological Context

The aquifer studied extends vertically from 20 to 130 m in depth counted from ground level and consists of karstified carbonates of the Middle Jurassic age. The aquifer is located at the border between the Paris (to the North) and the Aquitanian (to the South) sedimentary basins, along the so-called "Poitou doorstep" (Figure 1a). The HES covers an area of 12 hectares, its building phase started in 2002, and up to now, 35 wells have been drilled up to a depth of 120 m (Figure 1b). Hydrogeological investigations showed that the maximum pumping rates available varied highly from 5 to 150 $m^3$/h for similar full-penetrating wells of 20 cm in diameter at their bottom. The presence of pervasive karstic drains was witnessed by recent optic imaging of the well bores. Almost all the wells show small caves and conduits crosscut by the walls of the boreholes, with sometimes mean apertures of 0.2–0.5 m. These conduits are mostly concealed in three thin horizontal layers at 35, 88, and 110 m depth.

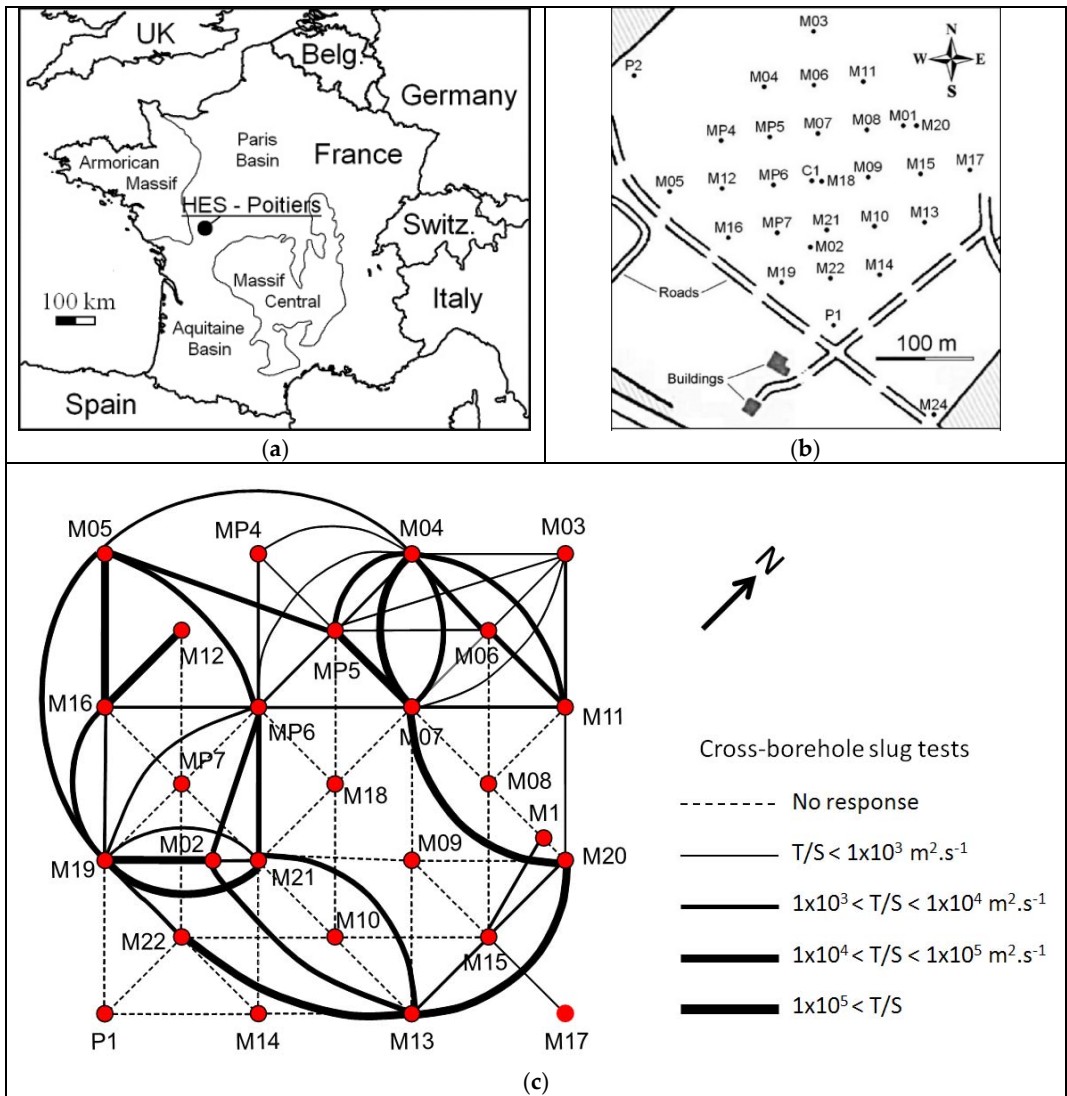

**Figure 1.** Hydrogeological experimental site in Poitiers. (**a**) Site map, (**b**) well location, (**c**) diffusivity map inferred from slug tests.

It is worth noting that these open layers are intercepted by both vertical fractures and now by the vertical wells which results in a good connection between the different layers and also between wells. Due to drilling works and various hydraulic tests, this connection is partly controlled by the degree with which drains were re-opened in the vicinity of the wells. Multiple hydraulic slug tests, usually rendering responses of the aquifer in the very close vicinity of the solicited well (a few meters around the well) show very rapid hydraulic pressure wave propagation, with interferences monitored in other wells distant of more than 100 m [11]. These observations allowed for mapping a hydraulic diffusivity distribution between pairs of wells, this diffusivity being the signature of the quality of connections between wells (Figure 1c) [11]. Finally, the highly varying connectivity between wells motivated further imaging of the subsurface regarding the location of possible water bearing bodies, or more generally open spaces prefiguring preferential water circulations. This imaging should both document the scale of a single well and the scale of an entire block of a few hundred meters on a side. High resolution geophysical tools seem to be well suited to undertaking that kind of investigation.

## 3. Imaging of a Near Surface Karstic Aquifer

The seismic method is conventionally used to give a 3D image of a reservoir. It covers the area of interest with regular sampling rates in three directions (inline, crossline and depth). The distribution of seismic velocities and the seismic frequency bandwidth fix the resolution over space, which is usually evaluated as the quarter of the dominant wavelength. In the HES field case, we will see that the 3D block is composed of cells, each with a volume in the order of 25 m$^3$. The 3D block has a sufficient resolution to detect three high-porosity karstic layers. However, the seismic resolution is not enough to detect karstic bodies of small size, but well measurement can be performed to try to observe these small bodies. Acoustic logging is the more efficient method, with a frequency bandwidth ranging between 1 and 20 kHz, and a vertical resolution on the order of 10 cm. Nevertheless, the lateral investigation in the close vicinity of the well is limited to a few tens of cm. In the HES field case, it will be shown later that acoustic logging detects karstic bodies that are not identified by the 3D seismic survey.

In addition to acoustic logging, Stoneley waves (or tube waves) observed on VSP can be used to detect highly permeable zones. When a downward-going P-wave reaches a permeable zone or a karstic body, the wave will move the concealed fluid. A conversion of downward propagating P-waves into downward and upward propagating Stoneley waves is then observed at depths occupied by highly permeable zones or karstic bodies. To favor the observation of wave conversion, a hydrophone can be used as the VSP tool. Two wells, C1 and MP7, are used to illustrate the acoustic logging and the specific P-Wave to Stoneley-wave conversion. In that sense, we show the interest of using both low resolution but large-scale investigation seismic method and very high resolution but small-scale investigation acoustic method to describe a reservoir at different scales.

### 3.1. 3D Seismic Imaging

Due to the size of the experimental field, the length of the seismic line cannot exceed 250 m in the in-line direction. In the crossline direction, the extension of the area does not exceed 300 m, resulting in the implementation of 20 receiver lines were implemented, with a 15 m lag-distance between adjacent lines. Figure 2a shows the map locating the seismic lines and the wells. In the acquisition of data, a 48-channel recorder was used. Each shot point is composed of 48 traces with a single geophone (10 Hz) per trace (Figure 2b,c). The seismic source is an explosive source (25 g). This relatively simple system of source-receivers helps to identify and pick the first arrival times of propagated seismic waves.

A 5 m distance between two adjacent geophones was selected to avoid any spatial aliasing. A direct shot and a reverse shot were recorded per receiver line. Figure 2b shows an example of both in-line direct and reverse shots. Three shot points in the crossline direction were fired at distances of 40, 50 and 60 m from the receiver line under consideration. Figure 2d shows an example of a crossline shot. The range of offsets (source-geophone distance) was selected to optimize the quality of the seismic image over a depth in the reservoir between 40 and 130 m. The minimum offset distance was chosen equal to 40 m to reduce the influence of the surface waves. The time sampling interval is 0.25 ms and the recording length is 0.5 s.

A VSP, recorded in well C1 (Figure 2c), was processed to obtain a time versus depth law and a velocity model. A 3D refraction seismic tomography [12] was carried out to map the irregular shape of the top of limestone reservoir, and to obtain static corrections and a velocity model of the clay layer overlying the reservoir.

The processing sequence [13,14] includes amplitude recovery, deconvolution, wave separation, static corrections and normal move-out corrections (using the VSP velocity model). Each shot point (both along the crossline direction and along the in-line direction) was processed independently to obtain a single-fold section with a sampling interval of 2.5 m (i.e., half the distance between two adjacent geophones) along the in-line direction. The processing of an in-line direct and reverse shot gather resulted in a single fold vertical section with an in-line extension of 240 m (indicated by a blue arrow on the location map of seismic lines, Figure 2a) while a cross line shot gather resulted in a single fold section with an in-line extension of 120 m (the red arrow on the location map of seismic lines,

Figure 2a). The single-fold sections of arrival times were merged together to build a 3D block, and times were converted into depth via the VSP time versus depth law inferred in the C1 well (Figure 2c). The width of the block along the in-line direction was 240 m, and that along the cross-line direction was 300 m. In the in-line direction, the reference zero in the horizontal direction indicated the location of the source line. The distance of reflecting points from this source line varied between −120 m and 120 m in the in-line direction, and the distance between two neighbor reflecting points was 2.5 m. In the cross-line direction, the distance between two reflecting points was 5 m.

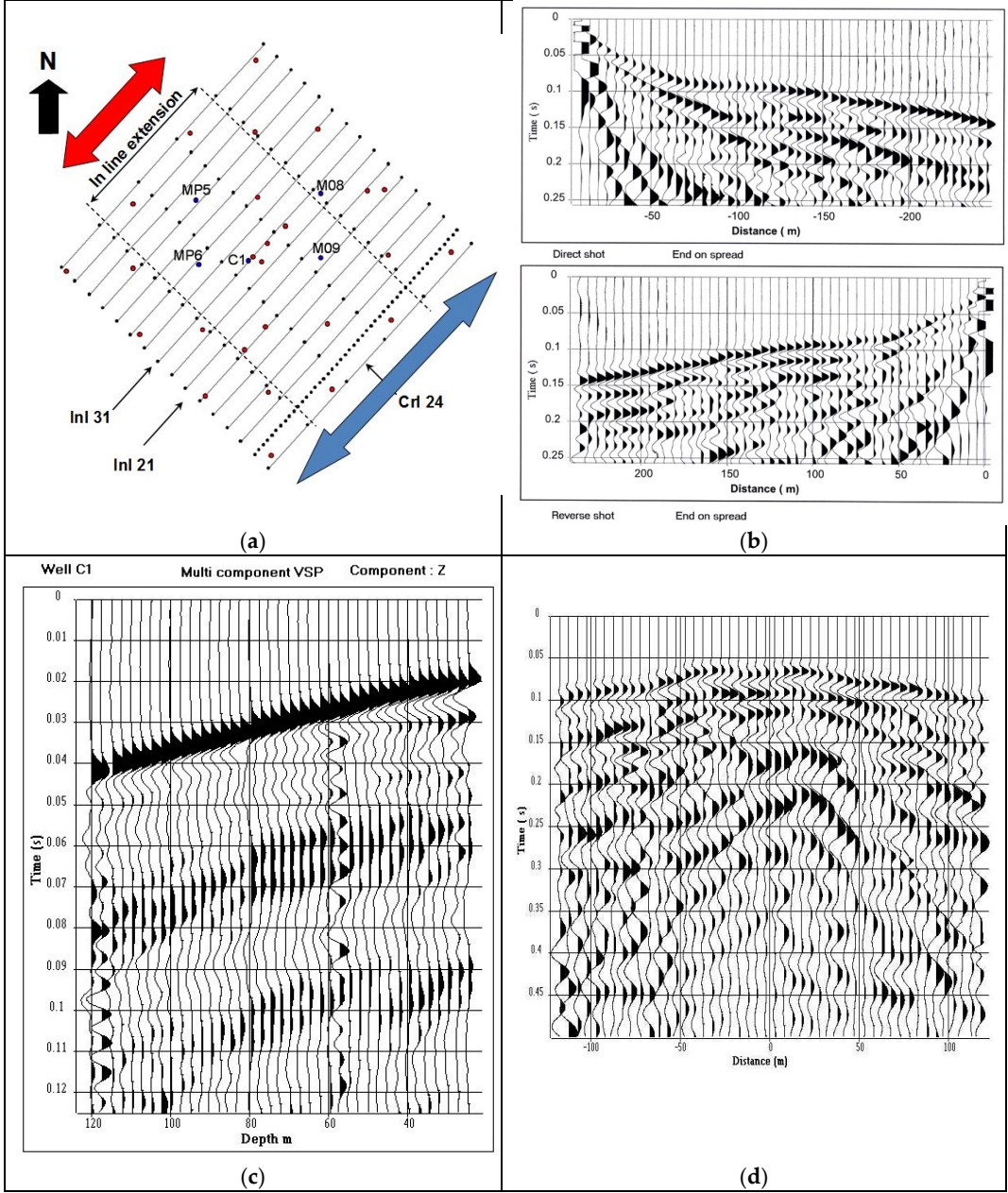

**Figure 2.** Three-dimensional seismic acquisition. (**a**) Seismic line implementation and well location (red points), (**b**) example of direct and reverse in-line shot points, (**c**) vertical seismic profile (VSP) at well C1, (**d**) example of crossline shot point.

The 3D amplitude block was converted into a 3D pseudo velocity block in depth, using velocity functions (with sonic logs recorded at wells C1, MP5, MP6, M08, M09) as constraints [13]. The pseudo velocity sections of the 3D block were then assembled with velocities obtained by refraction tomography

to create a 3D extended velocity field from the surface to the bottom of the aquifer. Figure 3a shows the results obtained for the in-line #31 seismic section of the 3D extended velocity field. It also shows the velocity map at 87 m depth in the reservoir (Figure 3b). The 3D velocity field reveals strong local heterogeneities of the aquifer along both the horizontal and vertical directions.

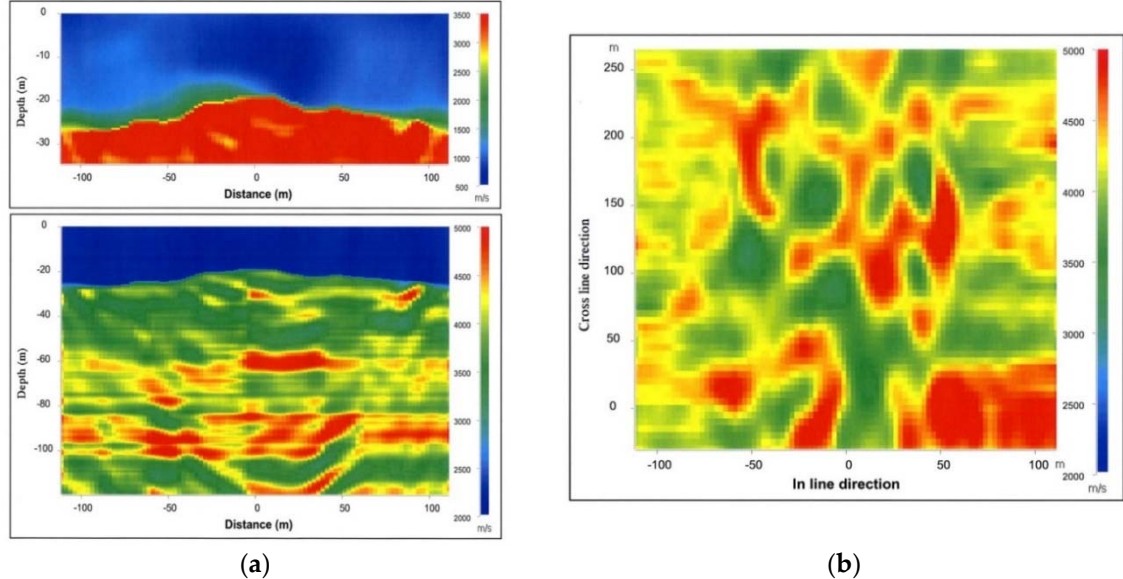

(a) (b)

**Figure 3.** Seismic processing, (**a**) inline #31 pseudo-velocity section (**upper part**: zoom in the 0–35 m depth interval) (**b**) pseudo velocity map at 87 m in depth.

In order to quantify the porosity variations within that aquifer, the pseudo velocities were first converted into resistivity values by relying upon the empirical Faust's law (relationship between seismic velocity and formation resistivity) [15]. Resistivity values were then converted into porosity values, by using Archie's law [16]. The complete procedure was reported in [14], but a short reminder can be useful. Faust (1953) established an empirical relationship between seismic velocity $V$ (ms$^{-1}$), depth $z$ (m), and electrical resistivity of the geological formation $R_t$ (ohm.m). For a formation of a given lithology, the velocity $V$ can be written as: $V = C(z R_t)^{1/b}$, with $C$ and $b$ scalar coefficients of the Faust's law. At each well where a complete log of electrical resistivity was available, an interval of velocity was extracted from the 3D seismic velocity block. The two sets of data consisting of resistivity log and local seismic velocity were combined to calibrate a local Faust's law, $V$ versus $R_t$ with optimal local $C$ and $b$ coefficients determined by least-squares. With a set of $C$ and $b$ coefficients at each well, these data were interpolated to form two 2D maps of the coefficients at the scale of the HES. Then, these interpolated values were used to transform the velocity 3D block into a block of equivalent resistivity $R_t$.

With a resistivity block available, then Archie's law was employed to transform the resistivity into porosity by following the relationship: $R_t/R_w = \phi^{-m}$, with $R_w$ (ohm.m) the water (concealed in the formation) electrical resistivity, $\phi$ the formation porosity, and $m$ a scalar exponent. $R_w$ was set to 20 ohm.m and the exponent $m$ to 2, the latter being a commonly reported value in the literature for well-cemented sedimentary formations. Notably, this classical value was adopted at the HES, mainly because the system remains a karsified continuous aquifer where karstic open volumes only represent 2–4% [11] of the total volume of the aquifer.

Figure 4 shows velocity and porosity vertical sections for the in-line #21 (Figure 4a) and for the crossline #24 (Figure 4b). Figure 4c shows the horizontal distribution of porosity values at 87 m depth and the location of the crossing wells.

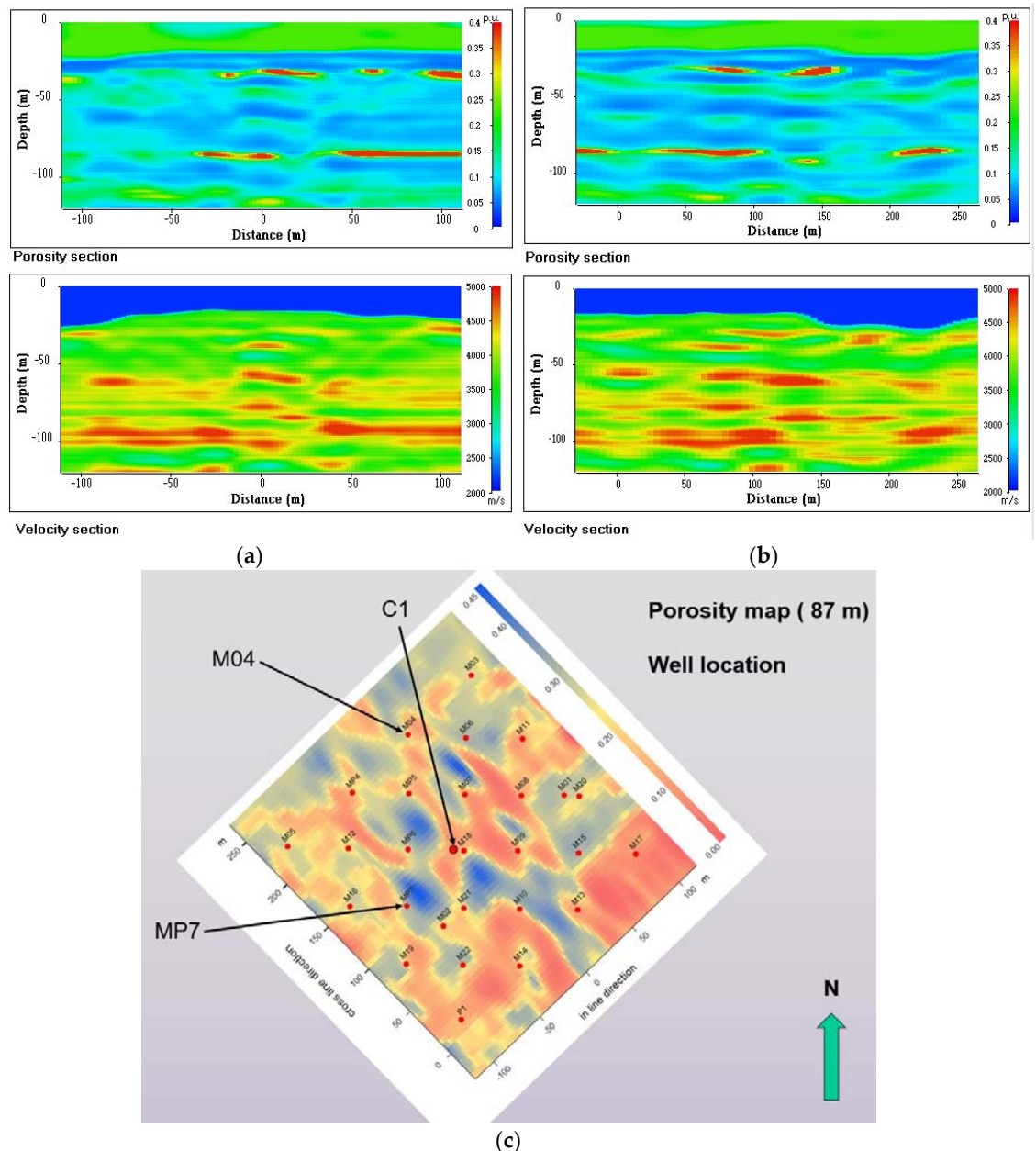

**Figure 4.** Velocity and porosity sections and porosity map (**a**) in line #21 section, (**b**) cross line #24 section, (**c**) porosity map at 87 m depth.

The resulting 3D seismic pseudo-porosity block reveals that three high-porosity, presumably-water-productive layers exist at depths of 30–40, 80–87 and 110–115 m (Figure 5a–c). The 80–87 m-depth layer is the most porous one, with some locations showing porosity values higher than 40%, which represent the most karstified layer of the reservoir. However, karstic heterogeneities only observed at wells C1, M04 and MP7 were detected at depths of 50–55 m depth interval. Figure 5d,e,f show the distribution of seismic velocity and porosity extracted from the 3D block at wells C1, M04 and MP7(see location Figure 4).

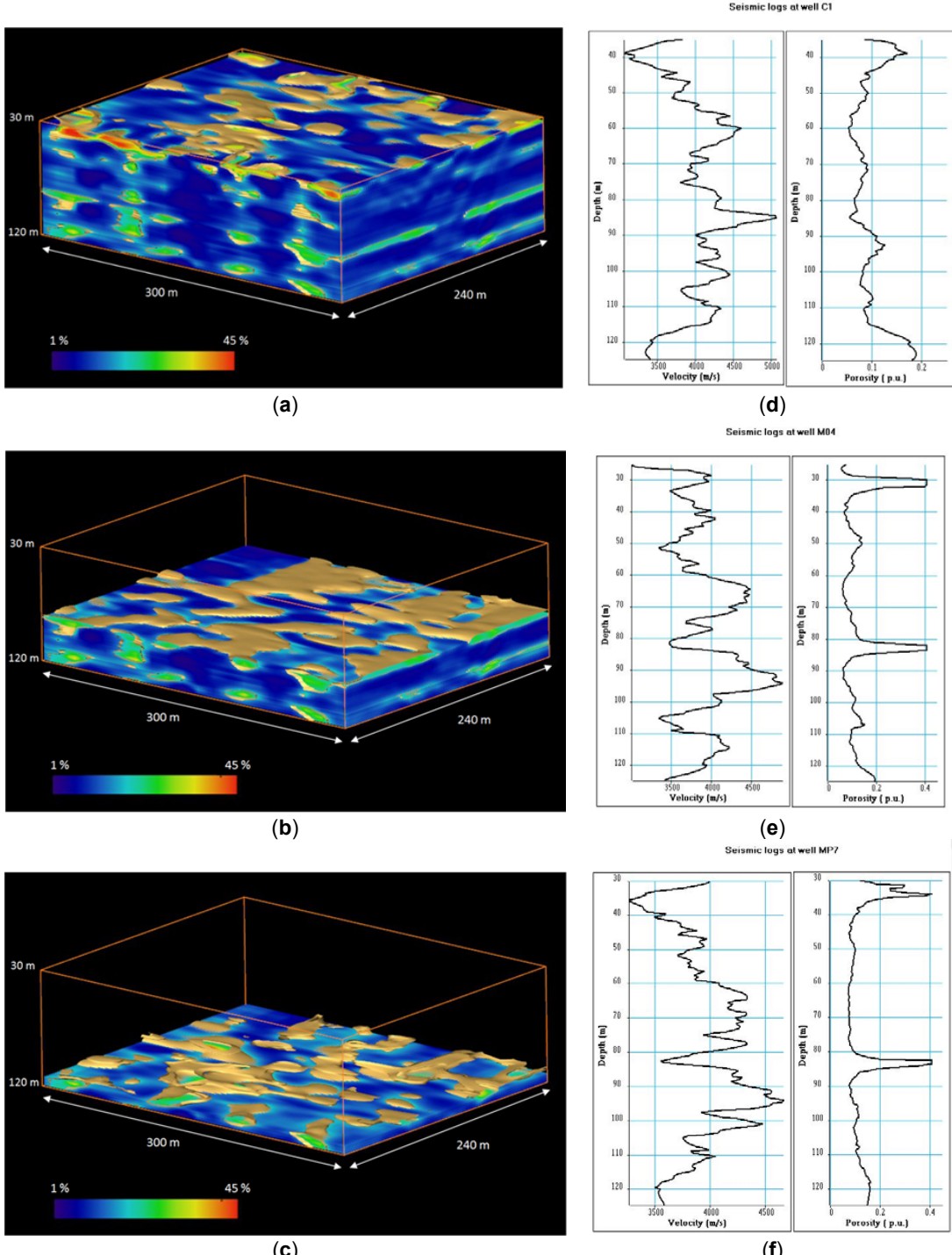

**Figure 5.** Distribution of karstic bodies from the 3D seismic block (**a**) at depth of 35–120, (**b**) 85–120, (**c**) 115–120 m. Seismic velocity and porosity vertical distribution at wells (**d**) C1, (**e**) M04, (**f**) MP7.

It is worth noting that the well C1 does not intercept karstic bodies in the 25–125 m depth interval but wells MP7 and M04 detects karstic features in the 30–40 and 80–87 m depth intervals. The seismic results at wells C1 and MP7 will be further compared with a full-wave form of acoustic data.

The 3D seismic block is composed of cells, each with a volume defined by $\Delta x = 2.5$ m along the inline direction, $\Delta y = 5$ m along the crossline direction, and $\Delta z = \lambda/4$ along the vertical direction, where $\lambda$ is the vertical wavelength ranging between 8 and 10 m. Consequently, a cell has a volume in the order of 25 m$^3$. The 3D seismic method can detect karstic bodies distributed horizontally over several

cells defined by high porosity values, as shown in Figure 5a,b,c. If a karstic body has a horizontal extension smaller than the horizontal extension of the cell, it will not be detected, but a small increase in the porosity over depth will be detected at the location of the karstic body, as observed at wells C1, M04 and MP7.

### 3.2. Borehole Televiewer and Seismic Data at Well M04

A borehole televiewer (BHTV) was run in well M04, over the 35–112 m depth interval. The analysis of the collected optic images shows that the borehole crosses several karstic bodies at different depths, mainly at depths: 31 (Figures 5e and 6a), 50–52 (Figures 5e and 6b), 83 (Figure 5e), and 106–108 m (Figures 5e and 6c).

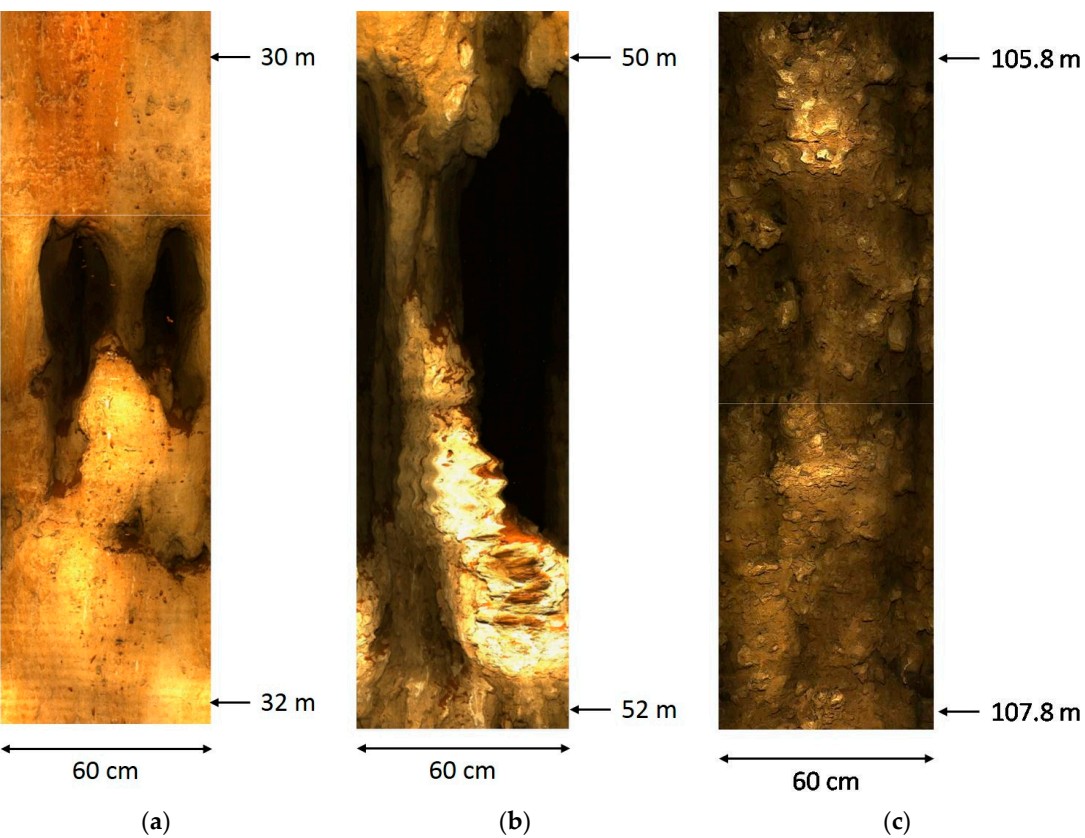

**Figure 6.** Borehole televiewer (BHTV) optic images at well M04 (**a**) at depth 31 m, (**b**) at depth 51 m, (**c**) at depth 107 m.

The levels at 51 and 107m have not been interpreted a priori by seismic imaging as karstic bodies. However, at these depths, significant increases in porosity are observed on the seismic porosity log (seismic porosity vertical distribution). The BHTV optic images confirm the presence of karstic bodies of small lateral extension.

The analysis of BHTV optic images shows that the aquifer is highly fractured. This led us to consider that the borehole conditions could introduce some risks to run logging tools in borehole M04. Consequently, no additional measurements, such as flow, acoustic or VSP data, were recorded in borehole M04.

### 3.3. Full Wave Acoustic Logging and Vertical Seismic Profile with Hydrophones at Well C1

Full waveform acoustic logging is based on the analysis and processing of the various wave trains (refracted, guided, and reflected waves) recorded by an acoustic tool. A geological formation can be simply defined by its acoustic parameters in the form of compression wave (P-wave) and shear wave

(S-wave) velocities, density, and factors of attenuation. With transmitters and receivers of the acoustic tool in the borehole, a part of the acoustic energy propagates in the fluid (or mud) concealed in the borehole. The most employed acoustic tools are monopole tools with multidirectional transmitters and receivers. In the fluid, transmitters generate a compression wave, which propagates in the formation as a P-wave and as a S-wave at the limit of refraction angles. In a vertical well, monopole tools allow for recording different propagation modes:

- the refracted compression wave;
- the converted refracted shear wave, only if the shear velocity of the formation is larger than the compression wave velocity in the mud;
- the fluid wave;
- the dispersive guided waves, that are pseudo-Rayleigh waves (only if the converted refracted shear wave exists), and Stoneley waves.

The acoustic probe for the logging of well C1 is a flexible monopole tool with two pairs of receivers: a pair of near receivers (1 and 1.25 m offsets) and a pair of far receivers (3 and 3.25 m offsets). The source is a magnetostrictive transducer. The receivers are independent, and each receiver has its own integrated acquisition device. The data were recorded through the far offset configuration. The sampling depth interval was 5 cm. The sampling time interval was 5 microseconds. The length of recording was 5 ms. The acoustic data are recorded in the 1–20 kHz frequency band. If the vertical resolution of the acoustic logging was very high (on the order of 10 cm), the lateral investigation close to the well was limited to a few tens of centimeters. The well C1 is equipped with a steel casing from the surface up to 22 m depth; below 22 m, it becomes an open-hole wellbore in which the water table was detected at 20 m depth during the logging.

Figure 7 shows the acoustic tool and the 3 m constant offset section (R1) in the 17–124 m depth interval. The vertical axis represents the depth from the ground level of receiver R1 and the horizontal one represents the recording time (ms). In the acoustic section, the refracted P-waves appear in the 0.5–1.0 ms time interval, the converted refracted S-waves and pseudo-Rayleigh waves in the 1.2–2 ms time interval, and the Stoneley wave in the 2–3 ms time interval. The logging is able to distinguish:

- the water table located at 20 m depth (no acoustic signal monitored above 20 m). Between 20 and 22 m, we observe some resonances due to a very poor cementation of the casing;
- the 22–33 m depth interval, where the signal to noise ratio is very poor, and only refracted P-wave are captured;
- the 33–60 m depth interval, in which the refracted P-waves, the converted refracted S-waves, and the Stoneley waves are visible. In the 49–54 m depth interval, all these waves are strongly attenuated;
- the 60–108 m depth interval corresponding with a very homogeneous profile with a very good signal to noise ratio. All the acoustic waves have a strong amplitude and are clearly visible;
- finally, the 108–124 m depth interval, which is also relatively homogeneous with refracted P-waves and Stoneley waves of good amplitude. The converted refracted S-wave is visible, but with a high level of noise.

The acoustic data were processed in the 35–124 m depth interval to obtain the velocity and the attenuation of the different wave trains. The arrival times of the different wave trains (refracted P-wave, converted refracted S-wave, Stoneley wave) were picked up by using the EarthQuick software which renders the sections reported in Figure 8 for the far (3 and 3.25 m) offsets. The curves in green and red correspond to the detected times of the different wave trains on the 3 and 3.25 m offset sections, respectively. The two sets of curves are displayed on each acoustic section.

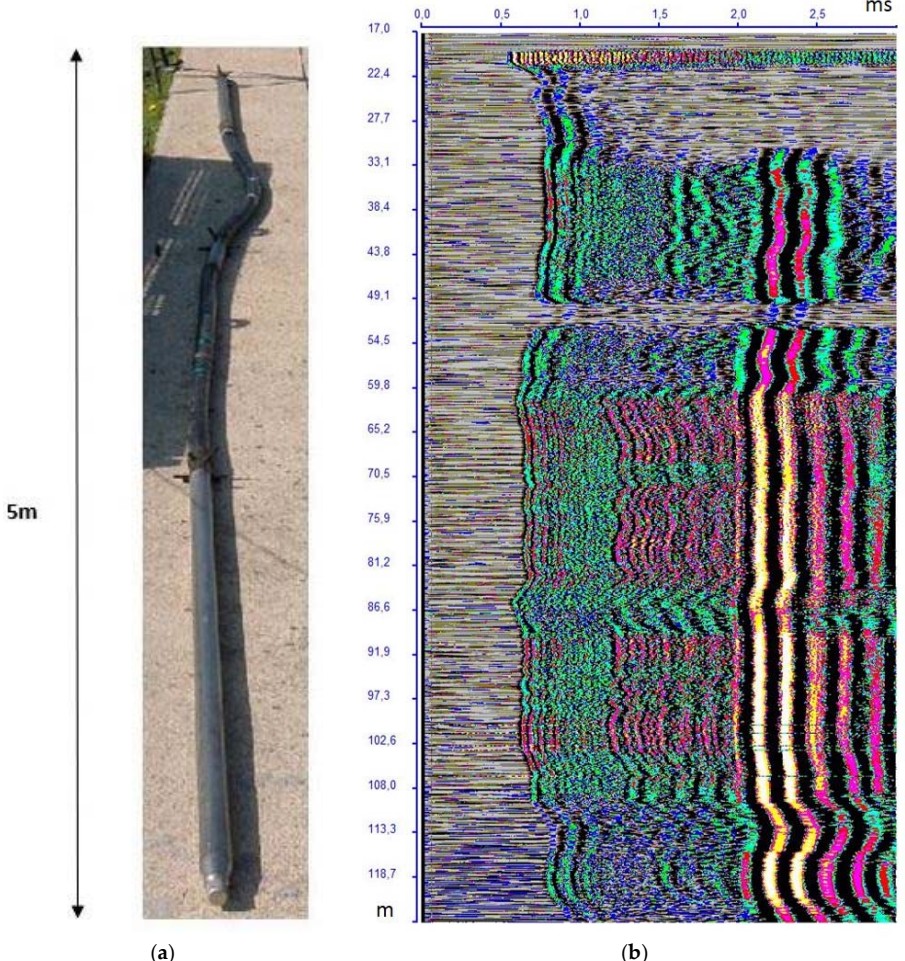

**Figure 7.** Full wave acoustic logging: (**a**) tool, (**b**) acoustic section (vertical axis: depth in m, horizontal axis: time in ms).

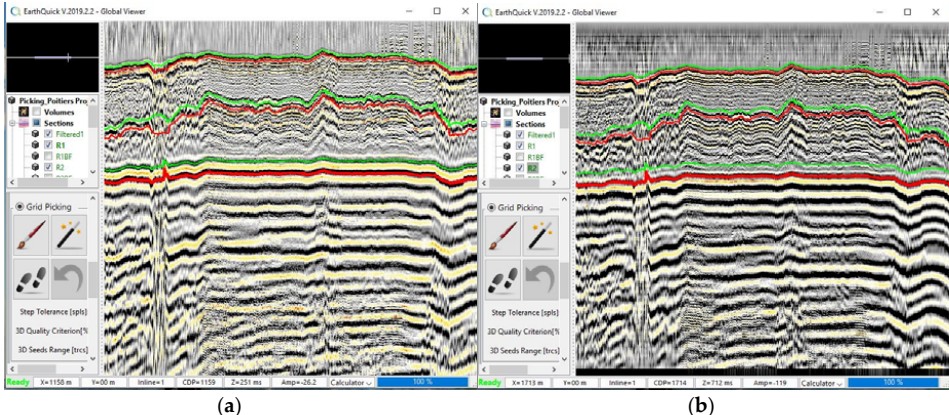

**Figure 8.** Screenshots of acoustic sections and picked times of the different wave trains by EarthQuick software. Vertical axis: time, horizontal axis: depth, (**a**) 3 m offset section (R1), (**b**) 3.25 m offset section (R2).

The detected times were used to compute the propagation velocity of the different waves. They were also used to define time windows for computing energy and attenuation logs.

Figure 9 shows the results obtained for the refracted P-wave. The acoustic sections R1 (3 m) and R2 (3.25 m) are muted, the acoustic signal before the arrival times of the refracted P-wave is zeroed. In a

250-µs window after the arrival times, the correlation coefficient between the two sections is computed at each depth in order to obtain the correlation coefficient log. For that purpose, the section R2 is time shifted to put in phase section R2 with section R1. The time shift value is the difference of travel times for the refracted P-wave observed between the two sections R2 and R1 at each depth. A high value of the correlation coefficient indicates that both the shape of the acoustic signal is the same on the two receivers (R1 and R2) and the picked times are accurate. A decrease in the correlation coefficient can indicate a poor picking due to either a poor signal to noise ratio or a change in the signal shape. The correlation coefficient log can be used to edit the velocity log. If the correlation coefficient is smaller than a given threshold value, the velocity value is cancelled and replaced par linear interpolation.

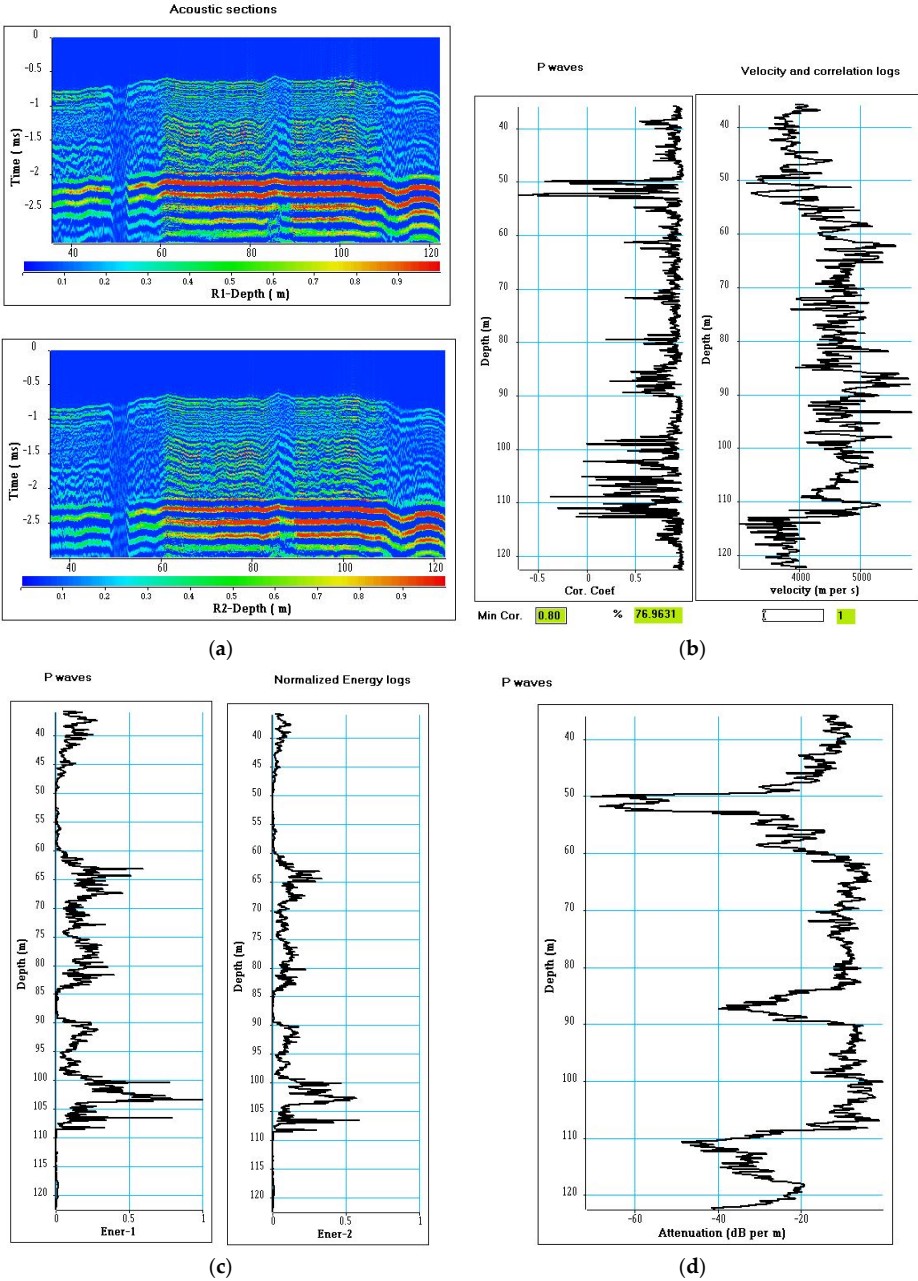

**Figure 9.** Refracted P-waves. (**a**) Acoustic sections after mute. Vertical axis: recording time in ms, horizontal axis: depth of receiver R1 or R2, color scale: normalized amplitude, (**b**) velocity log and its associated correlation coefficient log, (**c**) normalized energy logs, (**d**) attenuation log.

A threshold of 0.8 was introduced to edit the velocity log. More than 76% of the measured velocities were preserved. The time windows (250 µs) were also used to compute energy logs with the maximum of energy observed on the section R1 used to normalize the logs. The energy at a given depth is the stack of the square amplitudes of the wave over the window length. The attenuation log calculated for a vertical propagation of 0.25 m is simply defined as the ratio of energy logs at equivalent locations between the sections with the 3 and the 3.25 m offsets. This attenuation expressed in dB/m, is significant at three levels: 50–54, 85–90, and 110–120 m depth. The two deepest levels (85–90m and 110–120 m) are located in the karstic layers revealed by the 3D seismic imaging (Figure 5b,c). In the 50–54 m depth interval, a low energy and a very strong attenuation of more than 60 dB/m are observed.

The same procedure was applied to the converted refracted S-wave and the Stoneley waves. The results are shown in Figures 10 and 11, respectively.

For the converted refracted S-wave, the correlation coefficient is high in the 60–105 m depth interval, whereas the poor signal to noise ratio in the 35–60 m and 105–124 m depth intervals renders low correlation coefficients. Consequently, a threshold of 0.7 was selected. It preserves more than 53% of the measured velocities. However, the correlation coefficient between the two velocity logs (P-wave velocity shown in Figure 9 and S-wave velocity shown in Figure 10) is high (>0.77). The energy and attenuation logs are computed in 400 µs windows. In these windows (starting at the arrival time of the converted refracted S-wave), the wave packet is mainly of the pseudo-Rayleigh type, with elastic properties that mainly depend on the shear modulus of the rock.

For the Stoneley wave, a threshold of 0.9 preserves more than 90% of the measured velocities. The energy and attenuation logs were computed in 500 µs windows.

Irrespective of the type of wave (pseudo-Rayleigh or Stoneley), the same distributions of energy and attenuation over depth are observed. The energy and attenuation logs underline the same three levels identified by the analysis of the refracted P-wave. A very strong attenuation is measured in the 50–54 m depth interval, as attenuation measured in the two deepest levels (85–90 and 110–120 m) is weaker. The energies of the different wave packets depend on the permeability of the different ways, with the sensitivity to permeability changes partly evidenced by a loss in the total energy of the acoustic signal (P-waves, pseudo-Rayleigh waves, Stoneley waves) but mainly marked by the loss of energy of the pseudo-Rayleigh waves [17].

With a lateral distance investigated of a few centimeters, the Stoneley waves are very sensitive to the wall surfaces of the borehole. Usually, a loss of continuity of Stoneley waves indicates the presence of open fractures connected to the borehole, as the acoustic signature could indicate at 50–54 m depth. That being said, and in view of the geological context, this acoustic signature could also be associated with the presence of a karstic body.

Finally, the analysis of the acoustic data leads to the following conclusions:

- the attenuation of the different wave packets indicates that two deep levels (85–90 m and 110–115 m) have a high permeability. However, the continuity of the acoustic images suggests that no geological discontinuity, such as a karstic body, exists at these depths. This interpretation is confirmed by the 3D seismic block showing that borehole C1 does not cross karstic bodies in the 60–120 m depth interval (Figure 5d),
- the loss of continuity of acoustic images between 50 and 54 m and the very strong attenuation of the different waves, mainly the Stoneley waves, suggest that this level could be a karstic body with a very small lateral extension, as indicated by the seismic log (Figure 5d).

In order to confirm the interpretation given to the acoustic logs, additional measurements were carried out in borehole C1 by means of vertical profiles with hydrophones (Figure 12).

The attenuation of the total acoustic signal, computed from energy logs in a 2.5 ms window (for which the start time is given by the arrival picked time of the refracted P-wave), points out three levels of significant attenuation: 50–54 m, 85–90 m, and 110–115 m (Figure 13c). Irrespective of the type of wave, these three levels are detected by both the energy and attenuation logs (Figure 9c,d, Figure 10c,d, and Figure 11c,d).

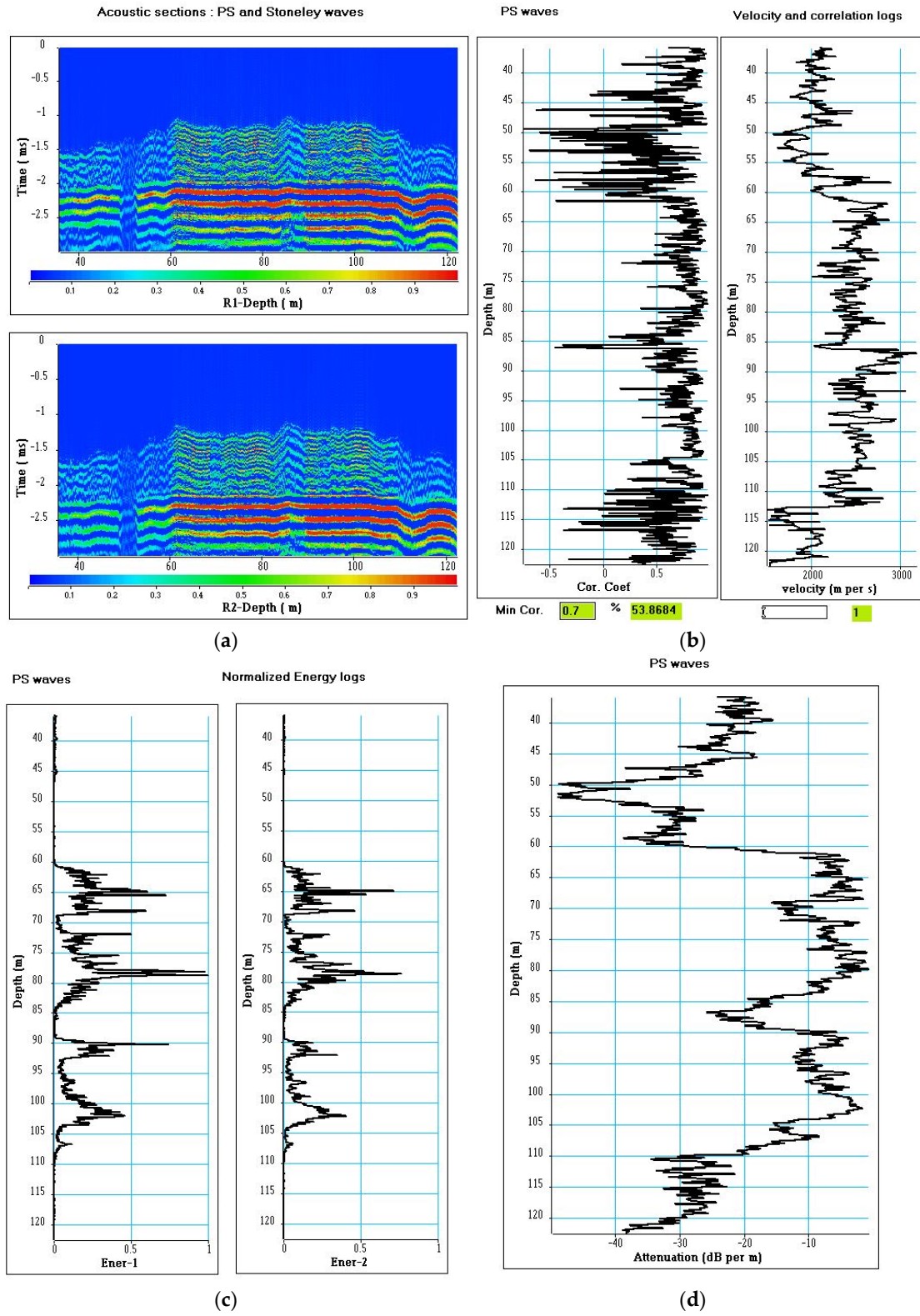

**Figure 10.** Converted refracted S-waves. (**a**) Acoustic sections after mute. Vertical axis: recording time in ms, horizontal axis: depth of receiver R1 or R2, color scale: normalized amplitude, (**b**) velocity log and its associated correlation coefficient log, (**c**) normalized energy logs, (**d**) attenuation log.

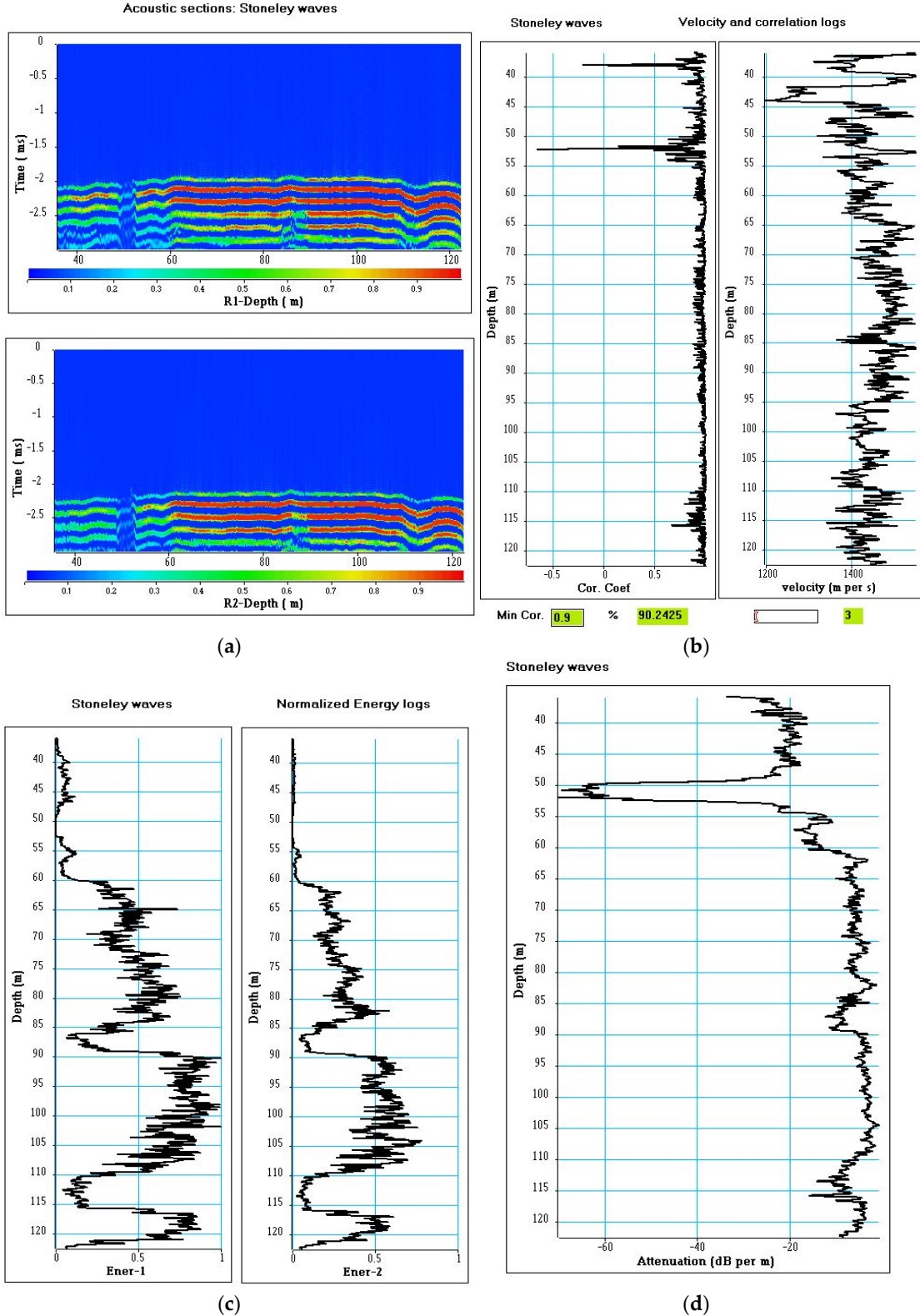

**Figure 11.** Stoneley waves. (**a**) Acoustic sections after mute. Vertical axis: recording time in ms, horizontal axis: depth of receiver R1 or R2, color scale: normalized amplitude, (**b**) velocity log and its associated correlation coefficient log, (**c**) normalized energy logs, (**d**) attenuation log.

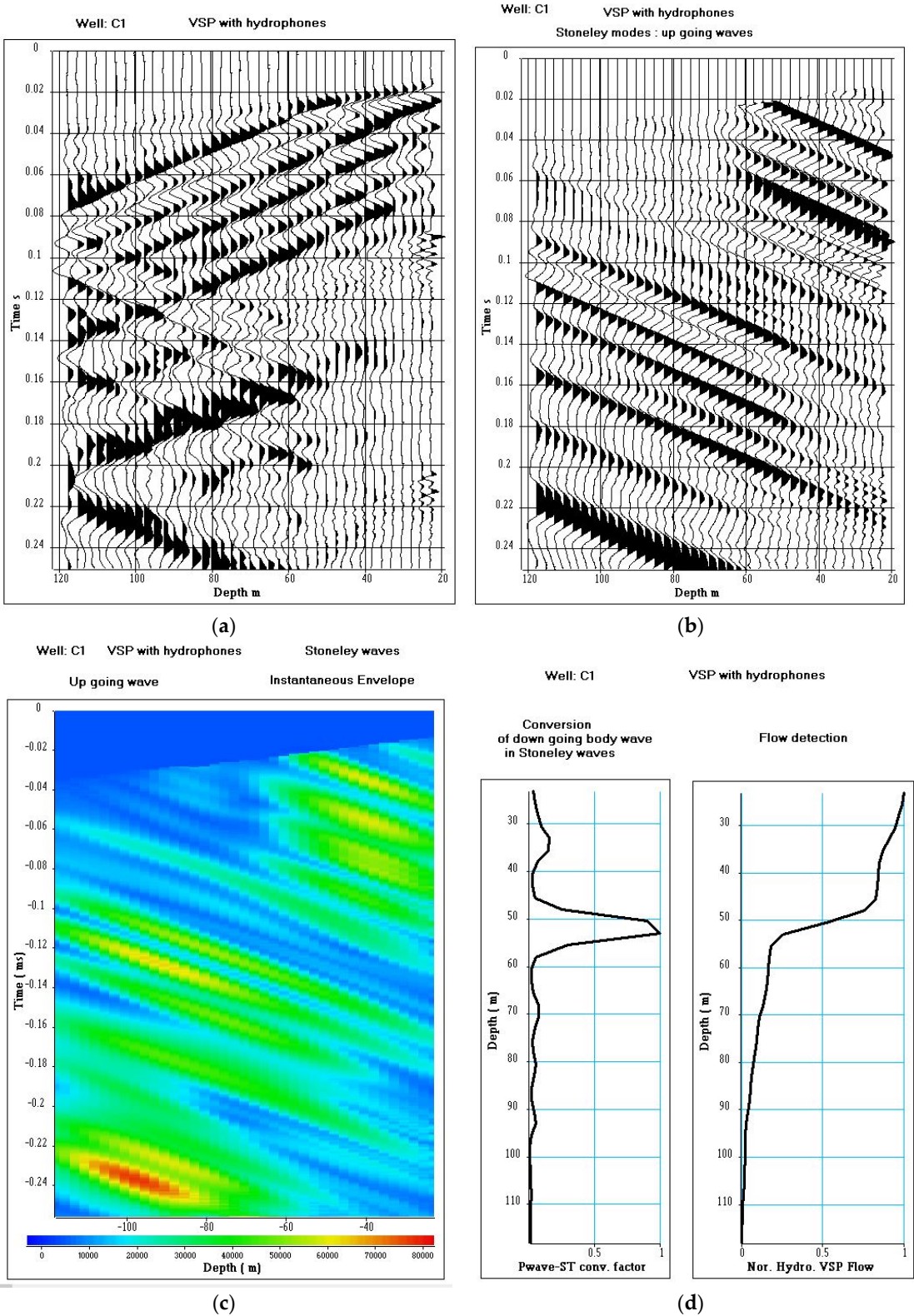

(**a**)　　　　　　　　　　　　　　　　　　　　　　(**b**)

(**c**)　　　　　　　　　　　　　　　　　　　　　　(**d**)

**Figure 12.** VSP processing. (**a**) VSP after amplitude recovery, (**b**) upward propagating Stoneley wave, (**c**) amplitude of upward propagating Stoneley wave, (**d**) P-wave/Stoneley-wave conversion factor and Normalized VSP flow.

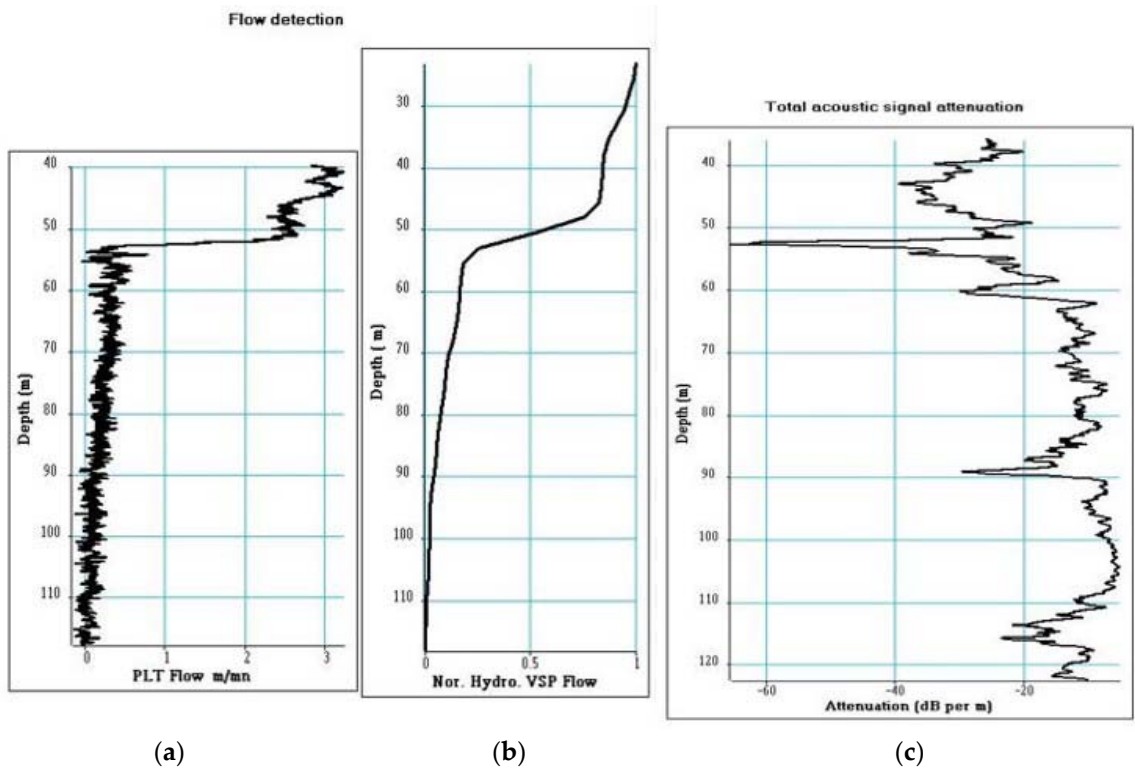

**Figure 13.** Comparison between (**a**) production logging tool (PLT) flow, (**b**) normalized VSP flow and (**c**) total acoustic signal attenuation.

For the VSP acquisition, the seismic source is a lightweight dropper and the borehole sensor is a hydrophone. The sampling step is 2.5 m for an acquisition performed over the 22.5–117.5 m depth interval. The VSP are highly corrupted by Stoneley waves. A conversion of downward propagating P-waves into downward and upward propagating Stoneley waves has been observed at the level of the karstic bodies. This phenomenon occurs in highly permeable formations [18,19] and can indicate the presence of flow to be confirmed by production logging tool (PLT) data [20].

Figure 12 shows the data recorded at well C1. We can notice the phenomenon of conversion from P- to Stoneley waves at 55 m depth (Figure 12a). The first arrival, which is the downward moving P-wave, is strongly attenuated at 55 m depth. The P-wave is partly converted to a downward moving Stoneley wave, which is then reflected at the bottom of the well. The VSP data were processed by using an apparent velocity filter to extract the downward-going and upward-going Stoneley waves [20]. The upward-going Stoneley waves are shown in Figure 12b. At 55 m depth, we can clearly see an upward-going Stoneley wave, created by conversion of the downward-going P-waves. The Hilbert transform applied to the different wave fields allows for estimating their amplitude (instantaneous envelope, Figure 12c). The instantaneous amplitudes of the upward-going Stoneley waves were stacked in a small corridor located after the arrival time of the downward-going P-wave, with the aim to obtain a P-wave to Stoneley wave conversion factor (Figure 12d) which indicates a karstic level between 50 and 55 m depth.

Assuming that the conversion factor is linearly proportional to water fluxes, it can be integrated over depth from bottom to top to mimic a flowmeter and compared with a PLT log.

The VSP flow (Figure 12d) is normalized. It has a low vertical resolution, the sampling interval over depth being 2.5 m. Figure 13 (a and b) shows a comparison between PLT and VSP flow logs.

Despite the difference in vertical resolutions between PLT and VSP flow logs (1 cm for the PLT, 2.5 m for the VSP), both profiles are in good agreement and confirm that there exists an active flowing karstic body at 52 m depth, as previously detected by the total acoustic signal attenuation log.

### 3.4. Full Wave Acoustic Logging and Vertical Seismic Profile with Hydrophones at Well MP7

The seismic log (Figure 5g) extracted from the 3D block at the location of borehole MP7 points out two karstic bodies located at 31–35 m depth and 81–84 m depth. The karstic bodies belong to the karstic levels at 30–40 m depth (Figure 5a) and 80–87 m (Figure 5b), respectively.

In borehole MP7, acoustic data were recorded using the acoustic tool shown in Figure 7, with the far offset configuration. The sampling depth interval was 2 cm. The sampling time interval was 5 microseconds. The length of recording was 5 ms. The MP7 borehole is steel-cased from the surface up to 37 m depth, and below 37 m, the wellbore is open-hole. During the logging, the water table was detected at 22 m depth.

Figure 14 shows the 3 m constant offset section (R1) in the 34–109 m depth interval. The acoustic section is corrupted by noise. On the acoustic section, the refracted P-waves appear in the 1.0–1.4 ms time interval. Locally, one can figure out the existence of converted refracted S-waves and pseudo-Rayleigh waves in the 1.5–2 ms time interval, and the Stoneley waves in the 2.3–3 ms time interval. Fluid waves of very high frequency are also observed in the 2.1–2.5 ms time interval. We can notice a loss of continuity of the refracted P-waves with respect to depth, highlighted by the presence of criss-cross events. The acoustic section can be decomposed as follows:

- Between 34 and 37 m, some resonances are observed due to a very poor cementation of the casing.
- In the 37–49 m depth interval, the signal to noise ratio is very poor and only the refracted P-waves and the fluid waves are visible.
- In the 49–58 m depth interval, the refracted P-waves are strongly attenuated. We mainly see fluid waves.
- The 59–77 m depth interval is homogeneous with a poor signal to noise ratio. The wave train is composed of refracted P-waves and fluid waves. Specifically, between 64 and 71 m depth, refracted P-waves, the converted refracted S-waves, and the Stoneley waves are visible.
- The 77–83 m depth interval mainly conceals fluid waves. The depth interval corresponds to the karstic level observed on the seismic log.
- In the 83–101 m depth interval, the signal to noise ratio is very poor and only refracted P-waves and the fluid waves are visible. Between 91 and 101 m depth, the fluid waves are attenuated. The level of noise is high.
- The 101–109 m depth interval is homogeneous and shows the complete wave field (refracted P-waves, converted refracted S-waves, Stoneley waves, and fluid waves). The level of noise is high.

The acoustic data were processed within the 37.5–109 m depth interval to extract the velocity and the attenuation of the refracted P-waves. The attenuation of the total wave field was measured. The velocity and attenuation of the fluid waves were also extracted to serve as a quality indicator in a whole profile plagued by the high level of noise.

The picked times of the refracted P-wave were used to compute the P-wave propagation velocity and the associated correlation coefficients for editing. Due to the high level of noise, a threshold of 0.6 for the P-wave correlation coefficient has been introduced to edit the velocity log. More than 77% of the measured velocities are preserved.

The picked times were also used to define time windows for computing attenuation logs, a short time window (350 μs) for P-wave attenuation and a large time window (3 ms) for total wave field attenuation. The velocity and attenuation logs are shown in Figure 15c,d.

The acoustic sections R1 (3 m) and R2 (3.25 m) was muted, the acoustic signal before the arrival times of the refracted P-waves was zeroed (Figure 15a). Furthermore, a running singular value decomposition (SVD) filter was applied to the acoustic sections to distinguish the signal part from the noise in the raw data. For the section R1, the signal section and the noise section are shown in Figure 15b. On the noise section, we can observe coherent slanted events (Figure 16). The slanted events, usually called criss-cross events, are refracted events reflected on the edges of geological

discontinuities. The criss-cross events are mainly observed at limits of the depth intervals 49–58 m and 77–83 m. The depth intervals are highlighted by a significant decrease in velocity, a strong attenuation, low values of correlation coefficient and a loss of continuity of the acoustic section. The depth interval 77–83 m corresponds to a karstic level (detected by the seismic log, Figure 5g) located at 81–84 m depth. The depth interval 49–58 m is probably a karstic body with a lateral extension smaller than the extension of a seismic cell.

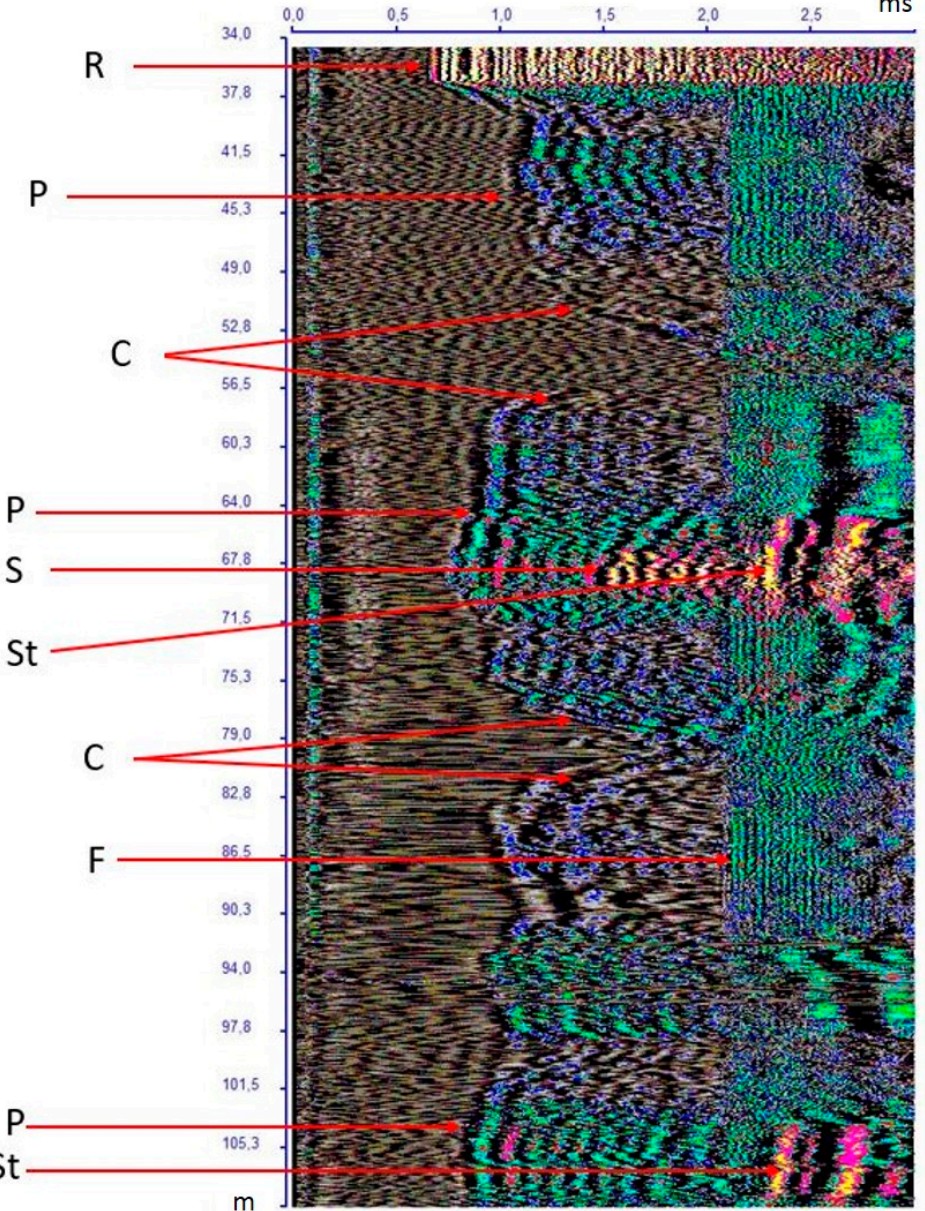

**Figure 14.** Acoustic section recorded in borehole MP7 (vertical axis: depth in m, horizontal axis: time in ms). P: refracted P-waves, S: converted refracted S-waves, St: Stoneley waves, F: fluid wave, C: criss-cross events, R: resonances due to the presence of a poorly cemented casing.

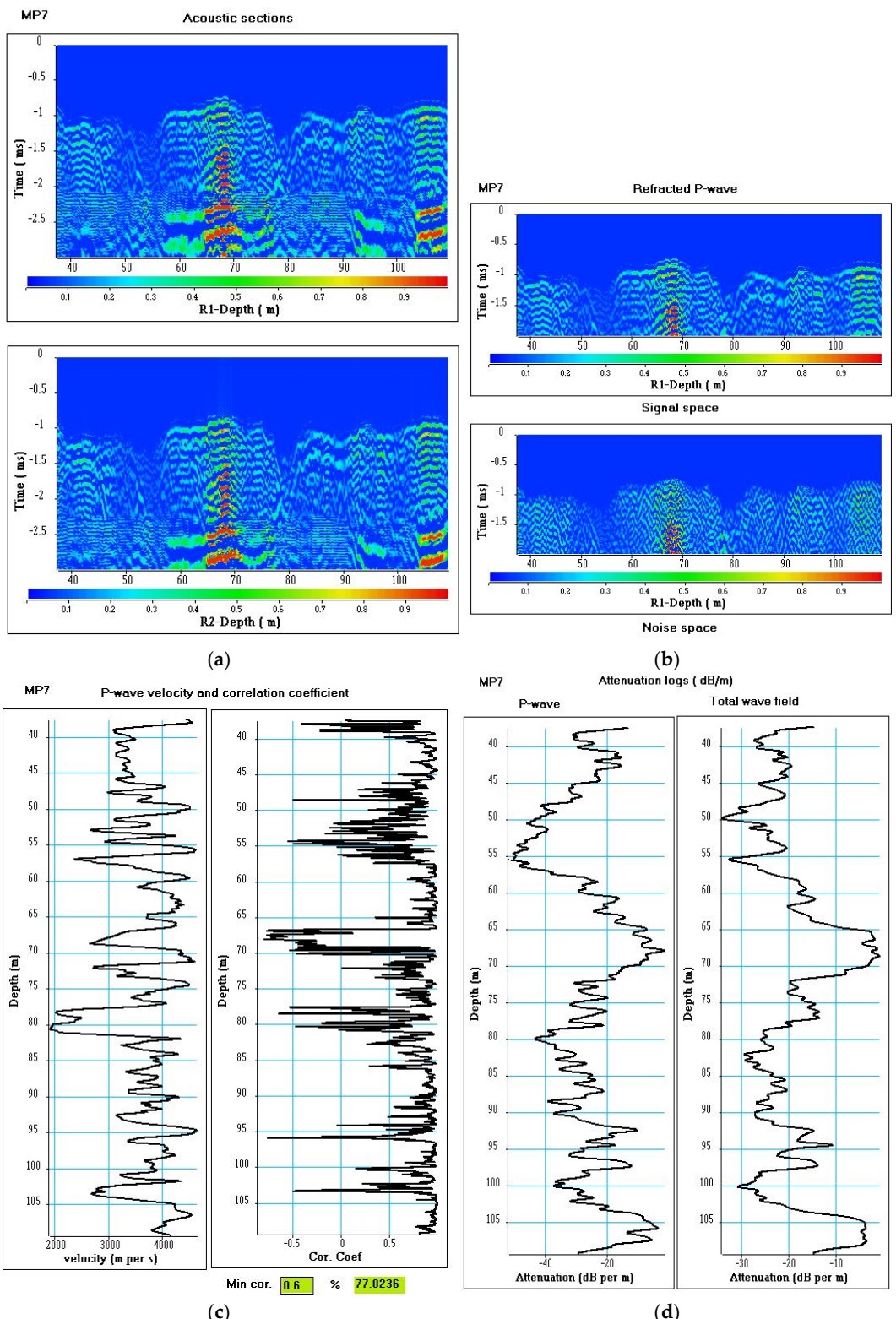

**Figure 15.** Refracted P-wave (**a**) acoustic sections after mute. Vertical axis: recording time in ms, horizontal axis: depth of receiver R1 or R2, color scale: normalized amplitude. (**b**) Signal and noise space sections. (**c**) P-wave velocity log and its associated correlation coefficient log, (**d**) P-wave attenuation log, total wave attenuation log.

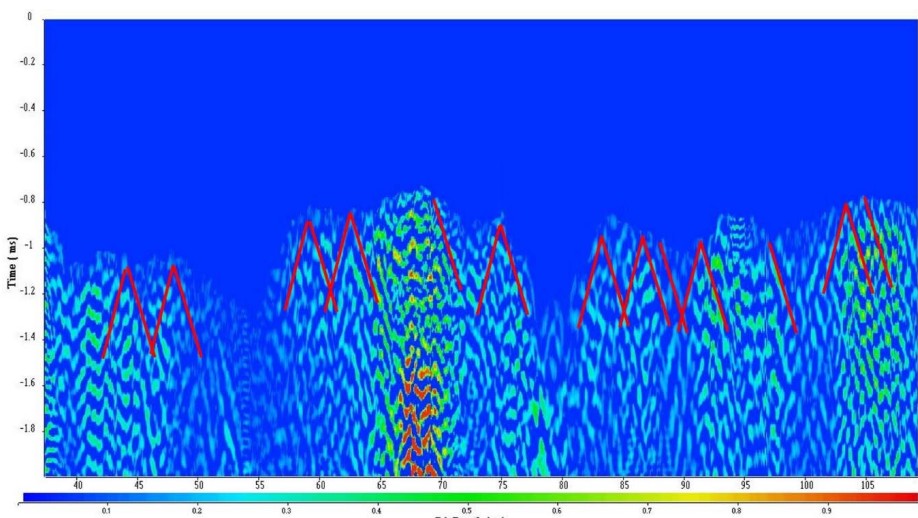

**Figure 16.** Focus on the noise space section presented in Figure 15b. The red lines highlight some criss-cross events.

The acoustic data were filtered in the 12–30 kHz frequency interval. The characteristics of the fluid waves were measured: velocity of propagation, correlation coefficient, and attenuation (Figure 17). The attenuation is weak, −4dB/min on average (with a standard deviation of 1.3 dB/m). The average fluid velocity is 1542 m/s (with a standard deviation of 17 m/s). A threshold of 0.6 for the fluid-wave correlation coefficient has been introduced to edit the velocity log. More than 72% of the measured velocities are preserved.

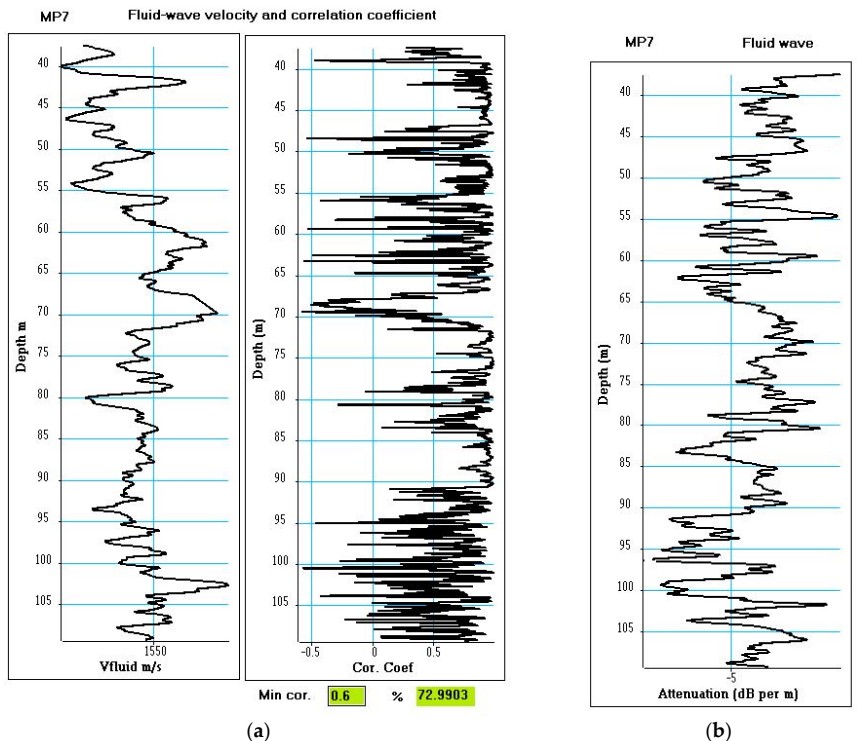

**Figure 17.** Fluid wave (**a**) velocity and correlation coefficient, (**b**) attenuation.

In order to confirm the interpretation, a VSP with hydrophones was recorded in borehole MP7. The same seismic source and borehole sensor as that for VSP in well C1 were employed. The sampling interval over depth is 2.5 m. The acquisition at well MP7 is done in the 42.5–112.5 m depth interval.

The VSP data (Figure 18a) were processed in order to extract the downward-going and upward-going Stoneley waves.

On the upgoing wave field (Figure 18b), we can clearly see conversions of the downward-going P-waves into upward-going Stoneley waves, at depths 57.5 and 82.5 m, as indicated by arrows. The blue arrow points out the karstic body, identified by the acoustic data in the depth interval 49–58 m. The red arrow points out the karstic body, identified both by the acoustic data and the seismic log (Figure 5g) in the depth interval 77–84 m. Three-dimensional seismic imaging, VSP data and acoustic logging lead to a consistent identification of karstic bodies, confirmed by BHTV optic images (Figure 19). The size of the karstic bodies varies between 10 and 1 m. On the HES, we observed that the maximum aperture of conduits can reach 2 m.

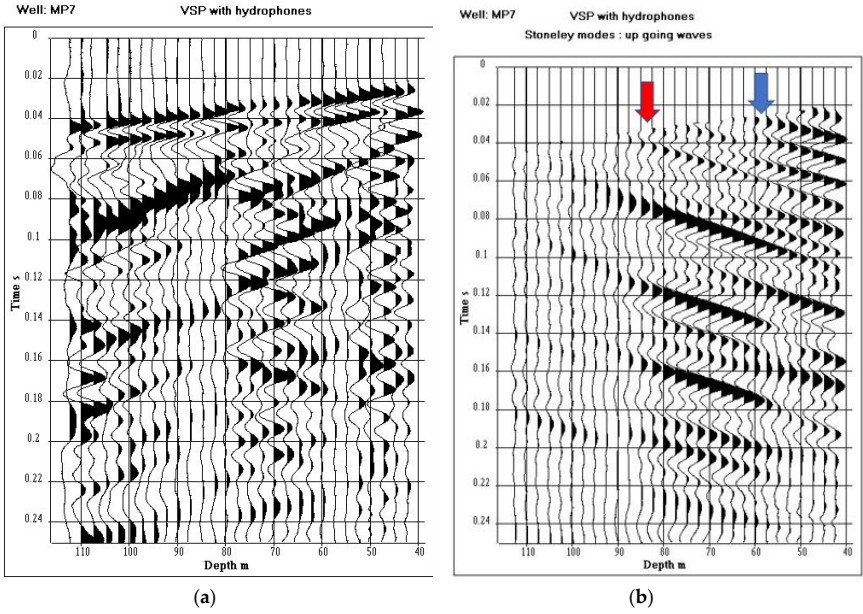

(a)  (b)

**Figure 18.** VSP processing. (**a**) VSP after amplitude recovery, (**b**) up-going Stoneley wave. Blue and red arrows indicate the position of karstic bodies detected by acoustic logging.

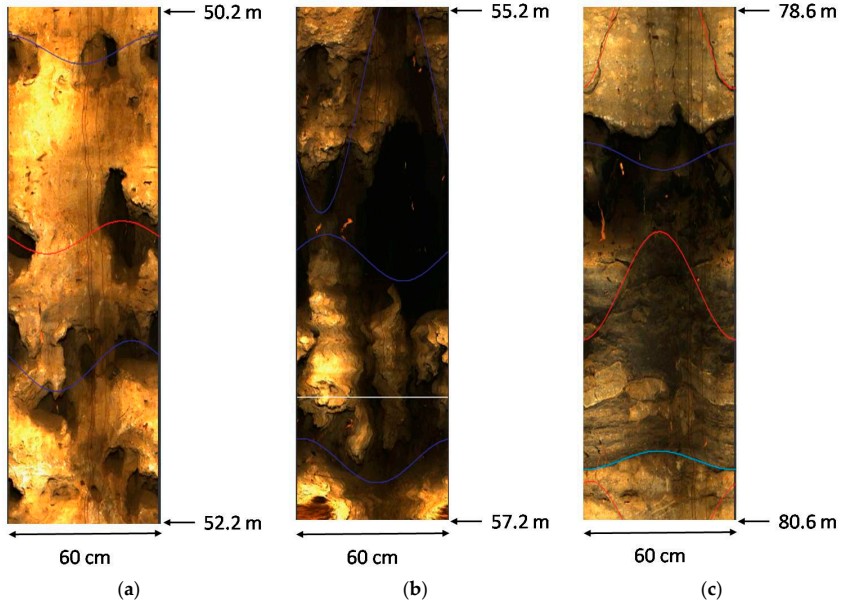

(a)  (b)  (c)

**Figure 19.** BHTV optic images at well MP7 (**a**) at depth 51 m, (**b**) at depth 56 m, (**c**) at depth 80 m.

## 4. Conclusions

The HES in Poitiers conducts experiments in a karstic aquifer of regional extension which is not a karst system in the common sense, where flow would only occur through wide open spaces of the near subsurface. In the HES, but also in many limestone and chalk aquifers made of tabular geological formations in sedimentary basins, flow is almost continuous over the whole system but with preferential layers or small subareas partly dissolved with increased porosity and hydraulic conductivity. Investigating these types of media via geophysical techniques assumes that methods sensitive to both large-scale and small-scale rock characteristics are employed to image the karstic features.

Three-dimensional high-resolution seismic imaging remains a cumbersome field experiment as it needs multiple shot lines and a dense network of receivers at the surface for rendering a valuable resolution in terms of elementary subsurface cells within which wave propagation velocities can be determined. In the HES, even with the dense network of shot lines deployed, this elementary cell is in the order of 25 $m^3$, which is a size to evidence karstic bodies, more precisely karstic layers, of large extension. An eventual application of the technique to probe a larger area than the $250 \times 300$ $m^2$ of surface area of the HES would probably lead to weaker resolution in the subsurface, and thus the capability to only evidence karstic layers of wide spatial extension.

If 3D seismic imaging can cover wide areas, it is also associated with the main downside that a velocity model is required to transform wave propagation characteristics in the aquifer into an indicator of "voids" within the subsurface. Irrespective of the type of transformation between the wave velocity and the void indicator, the 3D seismic survey needs local borehole (logs) measurements to build the basis of the transformation. It is obvious that the more borehole logs are available, the more accurate the transformation will be, especially in heterogeneous systems where the same wave velocity might have several origins. That being said, it is not stated here that a simple 3D velocity block would not be enough for inferring the occurrence of karstic features at the large scale. It is simply stated that borehole logs are clearly an added value to calibrate seismic data.

In the case of systems with several boreholes available, the FWAL is really worth a try. Acoustic wave monitoring needs for a rigorous signal processing because of the multiple type of waves recorded within a wellbore by receivers. The overall procedure for carrying out FWAL experiments is relatively simple and cheap, but the scale investigated does not exceed the very close vicinity of the probed well. The technique locally complements 3D seismic imaging in detecting small karstic bodies not revealed by large-scale seismic wave propagation. It also renders by-products such as "synthetic" flow logs that can be compared to actual ones, and information that allows for constraining classical loggings when the question is to identify flowing and non-flowing objects crosscut by a wellbore. In that sense, FWAL might also improve the conditioning of 3D seismic raw data on borehole logs to better image karstic bodies at the large scale.

Coupling 3D seismic imaging and FWAL is probably not a suitable combination to image karstic aquifers at the regional scale, to fix the ideas, over territories of several km of horizontal extension. However, as shown with the present study, the coupling produces images of sufficient resolution for the accurate delineation of karstic bodies at a scale compatible with the security perimeter usually surrounding subsurface catchments for water supply. In view of the short transit times associated with preferential flow paths in karstified aquifers, and the subsequent risks of rapid contamination of catchments, it makes sense to envision 3D seismic imaging and FWAL as valuable tools for assessing the effectiveness of such perimeters.

**Author Contributions:** Conceptualization, J.-L.M.; methodology, J.-L.M., G.P. and F.D.; software, J.-L.M.; validation, J.-L.M, G.P. and F.D.; investigation, J.-L.M., G.P. and F.D.; resources, J.-L.M., G.P. and F.D.; data curation, J.-L.M. and G.P.; writing—original draft preparation, J.L.M.; writing—review and editing, J.-L.M., G.P. and F.D.; visualization, J.-L.M.; project administration, J.-L.M. and G.P.; funding acquisition, J.-L.M. and G.P. All authors have read and agreed to the published version of the manuscript.

**Funding:** The authors acknowledge financial support from the European Union (ERDF) and "Région Nouvelle Aquitaine".

**Acknowledgments:** The experimentation and the collaboration between IFP School and the University of Poitiers benefited from the support of INSU-CNRS through the Environmental Research Observatory H +. We thank the University of Poitiers for permission to use the different data sets, recorded on the hydrogeological site of Poitiers. We thank Pierre Gaudiani for the acquisition of full-wave acoustic data. We thank Ludovic Peignard who gave us the opportunity of using the EarthQuick software (https://www.earth-quick.com) for the time picking of acoustic sections.

**Conflicts of Interest:** The authors declare no conflict of interest.

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
