# Peer review of "Contribution of Full Wave Acoustic Logging to the Detection and Prediction of Karstic Bodies"

_water, doi:10.3390/w12040948_

Round 1

Reviewer 1 Report

In the abstract, the authors presented a major focus of their study as showing the "importance of full wave acoustic logging (FWAL) for validating 3D seismic results within karst formations provided such karst formation has a lateral extension over several seismic cells". Does this imply a limitation of FWAL? That sounds counter intuitive to me and I was watching out for clarifications in the manuscript. This was not very clear.

What would be potential limitation of applying FWAL to a karst unit with limited lateral extension? How would such limits be defined?

These questions were not answered in the body of the manuscript.

Also, I recommend restructuring the introduction to provide more context for the current work detailing what is known already and the identified gaps that this research fills. In my opinion, this work also show case application of standard methodology. Hence, it will be worth it to give some background to the methods.

Is this geological context presented in lines 64 - 96 originally part of this work (i.e. original work of the authors)? or extracted from existing published (or unpublished) work by other authors? If it is from existing published work, I recommend summarizing it and providing adequate citation. If it was done as part of this study (which I guess it isn't), it should be stated expressly.

Combining the methods and result under the "field examples" section as it is currently makes the manuscript clumsy and difficult to follow. I suggest either splitting the method and result or at least giving different sub-heading for the different methods.

Author Response

Dear Reviewer.

     We thank you for your meaningful and fruitful comments. A revised version of our paper has been released, in which we added information clarifying the contributions of 3D seismic and full wave form of acoustic logging applied to the characterization of a near surface karstic aquifer.

Reviewer: Also, I recommend restructuring the introduction to provide more context for the current work detailing what is known already and the identified gaps that this research fills. In my opinion, this work also shows case application of standard methodology. Hence, it will be worth it to give some background to the methods.

In the revised manuscript, the Introduction has been modified to present more deeply the context of the study and the contribution of each geophysical method (3D seismic and FWAL) according to their resolution and the scale investigated. Section 3.1 is written to describe how 3D seismic is used to obtain a 3D image of the reservoir. The methodology for both the acquisition of rough seismic data and their transformation into “pseudo-porosity” blocks (high level of voids meaning here occurrences of karstic features) is detailed in the references 13 and 14 quoted in the manuscript. This notwithstanding, we extended the writing of Section 3.1, especially regarding the transformation of velocities into porosities. Notably, the background of the FWAL method is also better documented in Section 3.2 (see also reference 14).

Regarding the Hydrogeological Experimental site (HES) in Poitiers, 35 wells were drilled to perform hydrogeological experiments, such as slug tests, interference testing, etc. A BHTV (borehole televiewer) was carried out in each well, rendering optic imaging evidencing, locating and dimensioning occurrences of karstic bodies. In 2004, a high frequency band (up to 200 Hz) seismic 3D survey was recorded at the HES. The processing of the 3D data resulted in a 3D seismic pseudo-porosity block which revealed three high-porosity layers, at depths of 30-40, 85-87 and 110-115 m. The BHTV images confirmed the existence these three high-porosity layers detected at the seismic scale and interpreted in view of optic images as karstic layers.

Seismic resolution is usually estimated as the quarter of the dominant wavelength, ranging here between 8 and 10 m. However, the 3D seismic data have not enough resolution both in the vertical and horizontal directions to detect karstic bodies of small size, like those visible on optic borehole images at wells M04, MP7 and C1 in the 45-60 m depth interval.

In addition to the specific processing of high-resolution seismic, another goal of the paper is to show how FWAL (acoustic logging) can complement seismic data to detect karstic bodies of (very) small size, not probed by seismic because of their spatial resolution.

Two wells, C1 and MP7, are used to illustrate the information brought by FWAL. In that sense, we exemplify the interest of using both low resolution but large-scale investigation seismic method and high resolution but small-scale investigation acoustic method to describe a reservoir.

Reviewer: What would be potential limitation of applying FWAL to a karst unit with limited lateral extension? How would such limits be defined?

 Full wave acoustic logging (FWAL) is a very high-resolution technique. The frequency bandwidth is ranging between 1 and 20 kHz. The vertical resolution is on the order of 10 cm (a quarter of the dominant wavelength). However, the lateral investigation close to the well is limited to a few tens of cm. In the HES field case, we show that acoustic logging detects karstic bodies that are not detected by the 3D seismic. The acoustic criteria used to detect karstic bodies are based on the measurement of the acoustic energy and attenuation of the different wave trains (refracted P-wave, converted S-wave) that compose the acoustic signal. Another criterion is the loss of continuity along acoustic sections.

 In addition to acoustic logging, Stoneley waves (or tube waves) observed on Vertical Seismic Profile (VSP) can be used to detect highly permeable zones. When a downward going P-wave reaches a permeable zone or a karstic body, the wave will move the concealed fluid. A conversion of downward propagating P-waves into downward and upward propagating Stoneley waves is then observed at depths corresponding to the karstic bodies detected by the FWAL. To favor the observation of this wave conversion, a hydrophone can be used as a VSP tool. 

Reviewer: Is this geological context presented in lines 64 - 96 originally part of this work (i.e. original work of the authors)? or extracted from existing published (or unpublished) work by other authors? If it is from existing published work, I recommend summarizing it and providing adequate citation. If it was done as part of this study (which I guess it isn't), it should be stated expressly.

We do not fully agree with this recommendation, because we think that any contribution should be a self-containing paper, with a minimal information allowing for any reader to grasp the overall context of the study, but also, in the present case, the main hydrogeological characteristics that motivated a multi-technique geophysical investigation of the system. That being said, the Section on "geological context" was renamed "Hydrogeological context", the section was slightly reduced and only focused on the main hydrogeological characteristics that trigger the production of various imaging of karstic features riddling the HES system.

Reviewer: Combining the methods and result under the "field examples" section as it is currently makes the manuscript clumsy and difficult to follow. I suggest either splitting the method and result or at least giving different sub-heading for the different methods.

The Section “Field examples“ has been renamed as “Imaging of a near surface karstic aquifer” and a small paragraph has been added to introduce the different parts of the sections:

  • 3D seismic imaging
  • FWAL at well C1
  • FWAL at well MP7

In each sub-section named “FWAL at well C1” or “FWAL at well MP7”, we describe the acquisition and processing of acoustic data, we provide a qualitative interpretation of the acoustic section. We also show the different acoustic logs and their contribution to the karstic body detection. The contributions of 3D, FWAL and VSP are also discussed.

We hope that the revised manuscript gives you the answers to your comments, and we thank you again for your review and the time spent to help us improving the original manuscript.

Best regards,

Pr. Jean Luc Mari, for the authors.

Reviewer 2 Report

Dear Authors!

The paper is interesting and actual in the times when the society all over the world need to prospect for water reservoirs. Presented methods, especially advanced sonic log recording the full acoustic wavetrains are useful and effective. Processing of acoustic full wavetrains is explained in details. The presented simple idea of determining velocity of P-, S-, Stoneley waves by correlating wavetrains taking into account the depth shift because ot the receivers sites in the device, and assuming the level of cutoff/threshold for correlation coefficients as a source of quick and effective velocity determination is good. Similarly, energy loss calculated by comparing the second power of the amplitudes in the 250, 400 and 500 ms windows for P-, S- and Stoneley waves, respectively is quick and effective method.

In my opinion in the Discussion chapter it would be fruitful to select special part as Conclusions. Now, in this part of text you repeated the earlier descriptions. Presented results are interesting and well describrd and confirm the main idea of acoustic full wavetrains usefulness in quick interpretation supporting the seismic processing and interpretation.

Some remarks and suggestions to improve:

69 line – Cretaceous and tertiary eras – not eras but periods,

Line 147 – no information about the empirical constants in the Faust equation, are they adopted for the discussed area or they are just the same as Faust proposed,

Line 150 – resistivity was converted to porosity using Archie law, what was m coefficient, equal to 2 or other value adopted to carbonates in the study?

Fig. 5 Description of axes are too small – it is impossible to identify the karstic zone depth, description of colored scale is not visible,

Line 182 – what does mean seismic log, is it VSP?

Line 234 – EarthQuick software – not explained who is producer, is it free of charge and so on, please add some information.

Author Response

Dear Reviewer.

     We thank you for your meaningful and fruitful comments. We appreciated your comments regarding the procedure that we developed for processing acoustic data. A revised version of our paper has been released which clarifies the various contributions of 3D seismic data and full wave form acoustic logging to the characterization of a near surface karstic aquifer.

Regarding the Hydrogeological Experimental site (HES) in Poitiers, 35 wells were drilled to perform hydrogeological experiments, such as slug tests, interference testing, etc. A BHTV (borehole televiewer) was carried out in each well, rendering optic imaging evidencing, locating and dimensioning occurrences of karstic bodies. In 2004, a high frequency band (up to 200 Hz) seismic 3D survey was recorded at the HES. The processing of the 3D data resulted in a 3D seismic pseudo-porosity block which revealed three high-porosity layers, at depths of 30-40, 85-87 and 110-115 m. The BHTV images confirmed the existence these three high-porosity layers detected at the seismic scale and interpreted in view of optic images as karstic layers.

Seismic resolution is usually estimated as the quarter of the dominant wavelength, ranging here between 8 and 10 m. However, the 3D seismic data have not enough resolution both in the vertical and horizontal directions to detect karstic bodies of small size, like those visible on optic borehole images at wells M04, MP7 and C1 in the 45-60 m depth interval.

In addition to the specific processing of high-resolution seismic, another goal of the paper is to show how FWAL (acoustic logging) can complement seismic data to detect karstic bodies of (very) small size, not probed by seismic because of their spatial resolution.

Full wave acoustic logging (FWAL) is a very high-resolution technique. The frequency bandwidth is ranging between 1 and 20 kHz. The vertical resolution is on the order of 10 cm. However, the lateral investigation close to the well is limited to a few tens of cm. In the HES field case, we show that acoustic logging detects karstic bodies that are not detected by the 3D seismic. The acoustic criteria used to detect karstic bodies are based on the measurement of the acoustic energy and attenuation of the different wave trains (refracted P-wave, converted S-wave) that compose the acoustic signal. Another criterion is the loss of continuity along acoustic sections.

      Two wells C1 and MP7 are used to illustrate the points.

In the revised manuscript, the introduction has been modified to illustrate more clearly the context of the study and the contribution of each method (3D seismic and FWAL) according to their respective resolution and scale of investigation.

The Section “field examples “ has been renamed as “Imaging of a near surface karstic aquifer” and a small paragraph has been added to introduce the different parts of the sections:

  • 3D seismic imaging
  • FWAL at well C1
  • FWAL at well MP7

In each section named “FWAL at well C1” or “FWAL at well MP7”, we describe the acquisition and processing of acoustic data, we provide a qualitative interpretation of the acoustic section. We also show the different acoustic logs and their contribution to the karstic body detection. The contributions of 3D, FWAL and VSP are also discussed.

We hope that the revised manuscript gives you the answers to your comments, and we thank you again for your review and the time spent to help us improving the original manuscript.

Best regards,

Pr. Jean Luc Mari, for the authors.

Reviewer’s remarks and suggestions to improve:

69 line – Cretaceous and tertiary eras – not eras but periods:  OK

Line 147 – no information about the empirical constants in the Faust equation, are they adopted for the discussed area or they are just the same as Faust proposed

Faust (1953) has established an empirical relationship between seismic velocity V, depth z, and electrical resistivity measurements Rt. For a formation of a given lithology, the velocity V can be written as:

with C and b scalar coefficients of the Faust’s law.

At each well where a complete log of electrical resistivity has been recorded, an interval of velocity has been extracted from the 3D seismic velocity block. The two sets of data consisting of resistivity log and local seismic velocity were combined to calibrate an empirical Faust's law, which has then been used as a local constraining function to transform the 3D pseudo-velocity block into a 3D pseudo-resistivity. For each well, the two coefficients, C and b of the Faust’s law were determined through a least-square minimization of the difference between the 3D block-extracted seismic velocities and the velocities predicted from the Faust's law using the measured electrical resistivity log as input. 2D distribution maps of the C and b values over the site were interpolated and then used in the velocity - resistivity conversion procedure of the 3D seismic block.

Line 150 – resistivity was converted to porosity using Archie law, what was m coefficient, equal to 2 or other value adopted to carbonates in the study?

Archie (1942) empirically proposed that for water-saturated permeable formations, the relation between the true formation electrical resistivity, Rt, and the resistivity, Rw, of the water impregnating the geological formation was in the form:

where F is the so-called resistivity formation factor. f is proportional to the formation porosity and m is a “cementation” factor" as a characteristic of the geological formation. The F value derived from the resistivity measurement, Rt, is unaffected by the mineralogical components of the formation. Even though the cementation factor value may vary between 1.3 and 3 according to the lithology, an average value of to 2 is generally adopted for well-cemented sedimentary log. Even if the applicability of the Archie's law can be debated for a karstic reservoir, the reasons motivating its use are twofold. Firstly, the reservoir remains a continuous sedimentary carbonate formation, locally karstified (in opposition to a karstic system only made of wide open spaces), at the seismic resolution scale. The size of the elementary seismic horizontal pix (2.5 m along the in line direction, 5 m along the cross line direction), and the seismic vertical resolution ranging between 1 and 2 m, lead to an elementary seismic cell volume at least between 12 and 25 m3, a size for which the eventual karstic open space of the aquifer is small enough to consider the elementary seismic cell as a porous medium. Secondly, the volume of the karstic bodies only represents a few 2 to 4 % of the total volume of a reservoir, a fraction which was estimated by analyzing borehole images (Audouin, 2007)

For the above two reasons, the seismic-derived 3D resistivity block (Rt-seis) was converted into a 3-D pseudo-porosity block, by relying upon the Archie’s law with an exponent m=2, a water resistivity Rw=20 ohm.m and electrical resistivity Rt inherited from the block of Rt-seis.

Fig. 5 Description of axes are too small – it is impossible to identify the karstic zone depth, description of colored scale is not visible,

The figure has been replaced by a new one.

Line 182 – what does mean seismic log, is it VSP?

The 3-D block gives a distribution of porosity with a regular sampling (inline, crossline and depth). If the location of a well (M04, for example) is considered, one can extract from the 3D block the porosity distribution or a log of porosity at the seismic scale at the well location. The text has been modified as in the sense of:

The levels at 51 and 107m have not been interpreted a priori by seismic data as karstic bodies. However, at that depths, significant increases of porosity are observed on the log of seismic porosity (porosity as the transform of seismic local velocities along a vertical distribution). The BHTV optic images confirm the presence of karstic bodies of small lateral extension.

Line 234 – EarthQuick software – not explained who is producer, is it free of charge and so on, please add some information.

EarthQuick software has been developed by Ludovic Peignard who is the leading collaborator of a startup. The link (https://www.earth-quick.com) will provide the the reader with all the information concerning the software and the conditions for having a license.

The Earth Quick software has been developed for the interpretation of seismic sections. Seismic interpretation is mainly based on the picking of seismic geological horizons and discontinuities. In the paper, we use EarthQuick software for the picking of arrival times of the different acoustic waves by considering an acoustic section as a seismic section.

For reference to the software, we only added the link in the main text.
